



# Xylem water in riparian willow trees (*Salix alba*) reveals shallow sources of root water uptake by in situ monitoring of stable water isotopes

**Jessica Landgraf**[1,2]**, Dörthe Tetzlaff**[1,2,3]**, Maren Dubbert**[1,4]**, David Dubbert**[1]**, Aaron Smith**[1]**, and Chris Soulsby**[3]

[1]Department of Ecohydrology and Biogeochemistry, Leibniz Institute of Freshwater Ecology and Inland Fisheries, Müggelseedamm 310, 12587 Berlin, Germany
[2]Department of Geography, Humboldt-Universität zu Berlin, Rudower Chaussee 16, 12489 Berlin, Germany
[3]Northern Rivers Institute, University of Aberdeen, St. Mary's Building, Kings College, Old Aberdeen, AB24 3UE, UK
[4]Landscape Functioning, Leibniz Centre for Agricultural Landscape Research, Eberswalder Straße 84, 15374 Müncheberg, Germany

**Correspondence:** Jessica Landgraf (jessica.landgraf@igb-berlin.de)

**Abstract.** Root water uptake (RWU) is an important critical zone process, as plants can tap various water sources and transpire these back into the atmosphere. However, knowledge about the spatial and temporal dynamics of RWU and associated water sources at both high temporal resolution (e.g. daily) and over longer time periods (e.g. seasonal) is still limited. We used cavity ring-down spectroscopy (CRDS) for continuous in situ monitoring of stable water isotopes in soil and xylem water for two riparian willow (*Salix alba*) trees over the growing season (May to October) of 2020. This was complemented by isotopic sampling of local precipitation, groundwater, and stream water in order to help constrain the potential sources of RWU. A local eddy flux tower, together with sap flow monitoring, soil moisture measurements, and dendrometry, was also used to provide the hydroclimatic and ecohydrological contexts for in situ isotope monitoring. In addition, respective bulk and twig samples of soil water and xylem water were collected to corroborate the continuous in situ data. The monitoring period was characterised by frequent inputs of precipitation, interspersed by warm dry periods, which resulted in variable moisture storage in the upper 20 cm of the soil profile and dynamic isotope signatures. This variability was greatly damped at 40 cm, and the isotopic composition of the subsoil and groundwater was relatively stable. The isotopic composition and dynamics of xylem water were very similar to those of the upper soil, and analysis using a Bayesian mixing model inferred that overall $\sim 90\%$ of RWU was derived from the upper soil profile. However, while for the soil water signatures, the direct equilibrium method showed good comparability with in situ results, for xylem water, the cryogenic extractions signatures were only moderately or not at all comparable. Sap flow and dendrometry data indicated that soil water availability did not seriously limit transpiration during the study period, though it seemed that deeper ($> 40$ cm) soil water provided a higher proportion of RWU ($\sim 30\%$) in a drier period in the late summer. The study demonstrates the utility of prolonged real-time monitoring of natural stable isotope abundance in soil–vegetation systems, which has great potential for the further understanding of ecohydrological partitioning under changing hydroclimatic conditions.

## 1 Introduction

Plants – as the interface between atmospheric and soil water – have an important influence on the water cycle. In terrestrial ecosystems, transpiration from plants accounts for up to 90 % of evapotranspiration (Jasechko et al., 2013) and up to 70 % of incoming precipitation during the growing season (Kozii et al., 2020). Furthermore, vegetation intercepts and redistributes precipitation via canopy evaporation,

throughfall, and stemflow (Friesen and Van Stan, 2019). Subsequent interactions between soil particles and root water uptake (RWU) influence soil moisture and soil hydraulic conductivity which in turn affect infiltration and runoff (Thompson et al., 2010). As tree roots take up water, different species can have contrasting preferences in their source water pools (Jackson et al., 1995), while individuals of varying size and age are capable of taking up water from different soil depths (Kühnhammer et al., 2020). However, RWU cannot necessarily be simply correlated with root distribution as it also depends on moisture and nutrient availability at different times of the year (Ehleringer and Dawson, 1992). Consequently, there is still limited knowledge on the temporal dynamics of RWU and associated water sources, both at high temporal resolution (e.g. daily) and over longer time (e.g. seasonal) periods (Berry et al., 2018; Beyer et al., 2020).

Stable isotopes of water are convenient natural tracers commonly used for the estimation of sources and ages of runoff (McDonnell et al., 2010; Sprenger et al., 2019), partitioning of evapotranspiration (Williams et al., 2004; Rothfuss et al., 2010; Dubbert et al., 2014), and RWU depth distribution (Dawson and Ehleringer, 1991; Goldsmith et al., 2019; Beyer et al., 2018). With the development of compact laser spectrometry systems, in situ monitoring of stable water isotopes in soil and tree xylem is now facilitating higher temporal and spatial resolution assessment to advance well-established destructive sampling methods (Herbstritt et al., 2012; Rothfuss et al., 2013; Volkmann and Weiler, 2014; Oerter and Bowen, 2017). However, destructive sampling followed by cryogenic vacuum extraction is still widely used for stable water isotope analyses in plants (West et al., 2006; Yang et al., 2015; Orlowski et al., 2016; Sohel et al., 2021) and is usually required to corroborate in situ measurements (see Mennekes et al., 2021). Despite the complexity, daily maintenance, and resource demands, the application of in situ methods in different compartments of the critical zone including soil, trees (Kübert et al., 2020; Beyer et al., 2020; Marshall et al., 2020), and the atmospheric interface (Braden-Behrens et al., 2019) is increasing. Unfortunately, no general, widely tested setup for such in situ measurements has been established and agreed upon yet (Beyer et al., 2020; Marshall et al., 2020).

Despite limitations, destructive sampling in previous studies revealed important process-based insights into plant–soil–water interactions. For example, New Zealand riparian willows were analysed monthly over the course of 7 months showing that water sources fluctuated seasonally with RWU from the near-stream aquifer during summer and from groundwater during winter (Marttila et al., 2017). A labelling experiment of two small willow trees (*Salix viminalis*) in a lysimeter by Nehemy et al. (2021) also showed RWU to be variable and linked to soil water potential and tree water deficits, shifting deeper as the upper soil dried. In situ monitoring has also revealed other subtleties: sub-daily investigations of xylem water from several tree and shrub

species in French Guiana, China, and Germany have found morning RWU (when transpiration is low) to be sustained by deeper soil layers, while daytime RWU was concentrated in shallow soil layers (De Deurwaerder et al., 2020).

For xylem water sampling in particular, destructive methods still dominate the research field. However, it has been shown that the cryogenic extraction method can affect the results due to volatile organic compounds (like alcohols) which can be mixed into the extracted liquid sample (Martín-Gómez et al., 2015) as well as extraction of water held in cell walls (Barbeta et al., 2020). These effects on xylem water samples are only linked to cryogenic extraction and seem absent in in situ experiments. For example, Volkmann et al. (2016a) installed a probe horizontally inside the sapwood of a stem borehole to sample xylem water in situ. This method, while delivering a new approach for in situ measurements in plants, showed none of the above-mentioned effects but an unexplained $\delta^{18}$O offset. Marshall et al. (2020) suggested that this offset might be due to non-equilibrium conditions, leading to their new method of stem borehole equilibration, where the borehole went through the complete stem. This approach was tested to simplify the measuring method and allows for continuous evaporation of liquid xylem water into a flowing airstream that passes the borehole (Marshall et al., 2020). The model description of flow is based on a central core of moving air flow that passes a volume of still air, allowing water vapour to diffuse from and to the borehole wall or the moving air stream (Marshall et al., 2020). The method was successful for a cut stem, as well as a live, Scots Pine *Pinus sylvestris* (Marshall et al., 2020).

Despite the advantages, in situ xylem isotope monitoring in the field continues to be rare due to methodological and logistical challenges, including root distribution, soil heterogeneities, and ambiguous water sources. A recent in situ labelling experiment by Seeger and Weiler (2021) using the same type of stem borehole probes as Volkmann et al. (2016a) inside European beech (*Fagus sylvatica*) trees over the course of 12 weeks found that xylem water differed from actual RWU. This led to the conclusion that xylem water isotopic composition does not necessarily represent simple RWU but rather an integration over certain fractions of RWU from different sources in the past (Seeger and Weiler, 2021). They suggested sub-daily monitoring of tree xylem water isotopes might not be suitable to investigate short-term RWU dynamics and instead recommended focusing more on spatial heterogeneity in soil and xylem composition (Seeger and Weiler, 2021). Another labelling experiment by Mennekes et al. (2021) investigated plot-scale xylem water isotopes in three different trees of various species (*Pinus pinea*, *Alnus incana* and *Quercus suber*) at two different heights (15 and 150 cm) over the course of 10 weeks using similar tree probes described by Volkmann et al. (2016a). They sampled xylem water isotopes at ∼ 5-hourly intervals and showed that in situ measurements delivered more consistent results compared to destructive samples. Another experiment

from May to September 2018 applied an in situ system investigating $\delta^{18}$O in soil and xylem water of three beeches with a $\sim$ 2-hourly interval in Switzerland (Gessler et al., 2022). With a Bayesian isotope mixing model, they showed that the beeches did not compensate for restricted topsoil water by deeper uptake, though the trees recovered rapidly after the rewetting. Recently, a study by Kühnhammer et al. (2022) tested the novel in situ borehole equilibration method for xylem and roots of tropical trees, together with in situ soil measurements similar to Volkmann et al. (2016b). Over 4 months, they measured $\delta^2$H and $\delta^{18}$O in xylem, roots, and soil (the latter only at night and at least every second day) and conducted labelled irrigation events. In general, irrigation events were measurable, though single ones were not clearly distinguishable in xylem water isotopic compositions.

The integration of high-frequency measurements of water stable isotopes with other plant-physiological variables like sap flux and stem size variation can also increase our understanding of sub-daily and seasonal variations in xylem isotopic composition and improve the comparison of isotope data from different individual plants or trees (De Deurwaerder et al., 2020; Nehemy et al., 2021). Stem size variation is caused by the imbalance between canopy transpiration and RWU or its incremental increase (growth) (see Zweifel, 2016). Diurnal patterns can show a swelling (water uptake) or shrinking (transpiration) response of the tree, reflecting its stem water tension (Zweifel et al., 2005) or differences in osmotic water potentials depending on the sugar content of the phloem (Mencuccini et al., 2013). Sap flow monitoring, thus, also provides a continuous proxy of transpiration rates (Paloschi et al., 2021). Although reflecting the atmospheric vapour pressure deficit (Butz et al., 2018) and canopy conductance, sap flow rates may react to soil water availability as well, but these responses are highly species-specific (Brinkmann et al., 2016) and may depend on the distribution of individual water sources or access to groundwater (Süßel and Brüggemann, 2021). In general, low soil water content combined with high air temperatures may lead to moisture stress, resulting in lower leaf water potential and turgor, a decline in photosynthesis, stomata closing, decreased sap flow, and reduction in cell enlargement and growth (Joshi et al., 2016).

To identify water sources used by plants from isotopic composition, linear endmember mixing models like IsoSource (Phillips and Gregg, 2003) have been widely used (Barbeta and Penuelas, 2017). More recent approaches with Bayesian frameworks like SIAR (Parnell et al., 2010) or MixSIAR (Stock et al., 2018) also provide statistical uncertainty assessments (Rothfuss and Javaux, 2017). These models are based on the basic assumption that the endmembers include all potential sources of xylem water and are isotopically distinct (von Freyberg et al., 2020). Furthermore, all endmembers of a mixture should be identified, and uncertainties of the sampling/monitoring method should be taken into account. Recent findings suggest combining Bayesian mixing models with ecohydrological information like soil properties and climate to improve their results (Rothfuss and Javaux, 2017; Kühnhammer et al., 2020). Von Freyberg et al. (2020) stated that in situ monitoring of stable water isotopes with high temporal resolution may help detect isotopic anomalies (like fractionation or new water inputs) and, thus, result in more reliable water source attribution.

Here, we used continuous in situ monitoring of the natural abundance of stable water isotopes in soil and xylem water of riparian willow trees (*Salix alba*) under field conditions over a period of 5 (soil) and 3 (trees) months. Our overarching research question was as follows: can we generate new insights into fluxes across the soil–plant–atmosphere continuum from combined in situ isotope variation and conventional ecohydrological monitoring (e.g. sap flow, biomass accumulation, and soil moisture) to assess water sources tapped by the willow trees? In order to understand the uptake of potential water sources, we compared the isotopic composition of xylem water with soil water as well as precipitation, surface water, and groundwater from liquid sampling. We also used destructive bulk soil water and xylem sampling for comparison with in situ measurements. Furthermore, we also monitored soil moisture, sap flow velocity, stem size variation, and eddy flux covariance on site. Finally, we used these data in the Bayesian approach mixing model SIAR to calculate the likely distribution water sources used by the trees.

## 2 Study site

The study site is located in the SE of Berlin, Germany (Fig. 1a). The climate is continental temperate, with long-term (1981–2010) mean annual rainfall of $\sim$ 570 mm (Deutscher Wetterdienst, 2020a) and temperature of 9.3 to 10.0 °C (Deutscher Wetterdienst, 2020a). Berlin is situated in the North European Plain, which was formed during the Weichselian glaciation (STG, 2016). A glacier tongue stretching from east to west formed a bowl-shaped depression, which today contains the largest lake of the city, Lake Müggelsee, at an altitude of 32 m. The lake's inlet and outlet is the river Spree. As a result of groundwater pumping since 1905, lake water can infiltrate into the adjacent groundwater aquifer (Driescher et al., 1993). The upper unconfined aquifer at Lake Müggelsee consists of sandy and gravel sediments and is underlain by an aquitard of silt and till at about sea level (Driescher et al., 1993).

The Leibniz Institute of Freshwater Ecology and Inland Fisheries (IGB) is located (Fig. 1a) on the northern shore of Lake Müggelsee. In its grounds, an area of roughly 480 m$^2$ was chosen as our study site and focused on two willow trees (*Salix alba*). To the north of the trees is a small wetland and, in the east, a small woodland separated by a small stream (Fig. 1b). The stream is fed by artificial fish ponds, which are constantly sustained by pumped lake water. The two willow trees stand relatively isolated and were chosen to replicate

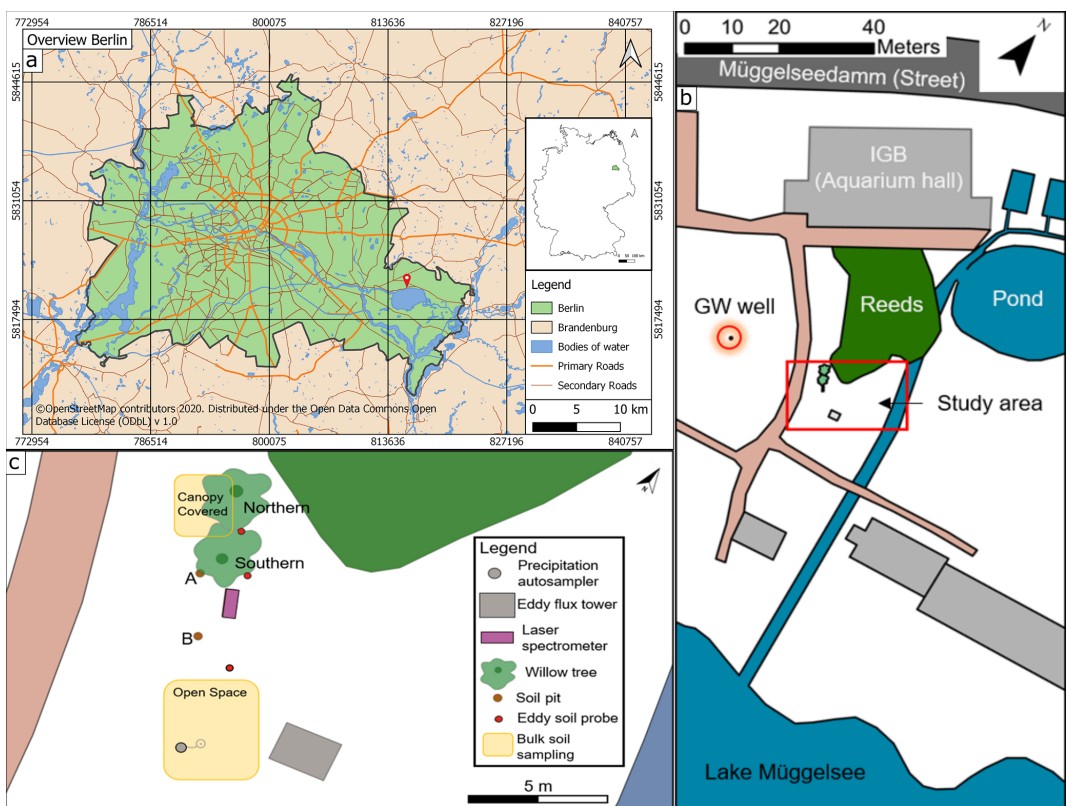

**Figure 1. (a)** Inner map: location of Berlin in Germany. Outer map: overview of Berlin, with red pin showing the location of the IGB (map data for map in A: © OpenStreetMap contributors, 2020). **(b)** Sketch of the study site (highlighted by red rectangle) with groundwater (GW) well (highlighted by red circle). **(c)** Bird's eye view showing all measuring points.

the same species and have similar ages (14 years). The northern willow (height $\sim$ 18 m, later referred to as Northern Willow) had a stem diameter of 398 mm and the southern willow (height $\sim$ 19 m, later referred to as Southern Willow) a stem diameter of 353 mm (16 July 2020). Willow trees were described by Nehemy et al. (2021) as having a fast resource acquisition strategy and are known to have large water demands due to typically high transpiration rates ($\sim$ 30 to 100 L d$^{-1}$; see Schaeffer et al., 2000). Hence, it could be expected that the trees respond rapidly to variations in water availability. In addition, willows are tolerant of waterlogged conditions and can take up water from groundwater, so they potentially have flexible strategies of RWU depending on soil moisture conditions (Marttila et al., 2017). Our site also includes the lake-fed stream to the east of the trees.

In addition to the willows and a few other trees fringing the stream and wetland, the area is covered by grass and moss. There is also a well, which we used to monitor groundwater level and isotopes. Lake Müggelsee is roughly 50 m away from the study site. Despite developing brown earth characteristics, the soils comprise relatively homogenous silty sands, with the upper soil layer having a higher organic content. Those soils developed on backfilled ground (following

construction work at IGB), which results in the topsoil being classified as an Anthrosol (Rossiter, 2007).

## 3 Data and methods

### 3.1 Hydroclimatic and hydrometric monitoring

The study was conducted over most of the growing season in 2020 (from 20 May to 11 October). Climate conditions were monitored using a weather station integrated into a portable eddy covariance system (Li-cor Biosciences, Lincoln, NE, USA, with LI 7500DS open-path analyser; wind measurements via Gill Windmaster pro and a Smart Flux 3 system, frequency 10 to 20 Hz, Burba, 2013) (Fig. 1c). Automatically calculated mean values were logged over 30 min. Measured air temperature, wind (speed and direction), vapour pressure deficit, solar radiation, precipitation, topsoil heat flux, and soil moisture were used in this study. To calculate soil heat flux and moisture, soil water potential was measured with ThetaSondes ML2 (Delta-T Devices, Burwell, Cambridge, UK; accuracy $\pm 2\%$ to 5%) at a depth of 5 cm. The eddy flux system automatically calculated evapotranspiration from the latent heat flux (Burba, 2013). However, as the eddy flux tower was not above the tree canopy, evapotranspiration data

were used mainly for qualitative assessment of dynamics rather than the absolute values. Further, the measurements of the eddy covariance were considered to reflect the general environment of the willows and not the specific site. In some cases (e.g. power outage), occasional data gaps occurred which were infilled with data from the IGB weather station, ca. 15 m away at the rooftop of the IGB building. The precipitation data of both stations were compared with the open-access precipitation data from the German Weather Service (DWD) of the Berlin-Marzahn station (Deutscher Wetterdienst, 2020b).

Soil moisture monitoring took place in two soil pits (Pit A and Pit B) at three depths: 10, 40, and 100 cm. Pit A was located next to the southern willow tree (Southern Willow) and partially covered by its outer branches, while Pit B was further south and not covered (Fig. 1c). Soil moisture and temperature were measured at the three depths with water content reflectometers CS616 (Campbell Scientific, Inc. Logan, UT USA; accuracy $\pm 2.5$ % for volumetric water content (VWC)) and BetaTherm 100K6A1IA thermistors T107 (Campbell Scientific, Inc. Logan, USA; tolerance $\pm 0.2$ °C (over 0 to 50 °C)), respectively with a CR800 data logger and multiplexer (Campbell Scientific, Inc. Logan, USA) logging every 10 min. A delay in completion of Pit A was related to COVID-19 lockdowns and resulted in a shorter period of record.

Groundwater levels nearby were monitored with an automatic data logger (groundwater level probe) at an interval of 15 min (see location in Fig. 1b). For comparison, the groundwater level was also measured manually once a week with a water level meter. The average groundwater level is around 2.2 m below ground level and relatively stable, with seasonal variations < 10 cm.

## 3.2 Ecohydrological monitoring

We continually monitored sap flow and variation in the stem circumference of the willow trees. Sap flow was measured at 15 min intervals using the heat ratio method (Burgess et al., 2001) with four sap flow meters (SFM1 instrument, ICT International, Australia), from which two per tree were installed on the north and south side of the trunk. All sensors were installed at breast height. The measured increase in sapwood temperature following the release of a heat pulse downstream and upstream of the heater is calculated into heat pulse velocity ($V_h$) according to Marshall (1958) as

$$V_h = \frac{k}{x} \ln\left(\frac{v_1}{v_2}\right) 3600,$$  (1)

where $k$ is the thermal diffusivity of fresh wood, $x$ is the distance between the heater and either temperature probe, and $v_1$ and $v_2$ reflect the increase in temperature at equidistant points downstream and upstream, respectively. Further information on the theory may be found in Burgess et al. (2001).

Variation in stem circumference was observed with two dendrometers (DR Radius Dendrometer, Ecomatik, Dachau, Germany), one per tree at the northern stem side, at a height of 95 cm measured from the soil surface. Data were logged at a 15 min interval with a CR300 data logger (Campbell Scientific, Inc. Logan, USA).

## 3.3 Stable water isotope monitoring

Stable isotopes of water were monitored in precipitation, groundwater, lake and stream water, bulk soil water, soil water vapour, and xylem water vapour. Precipitation water was collected with an ISCO 3700 autosampler (Teledyne Isco, Lincoln, USA) (see Fig. 1c). The autosampler bottles were filled with a paraffin oil layer > 0.5 cm in thickness (according to IAEA/GNIP, 2014) to avoid evaporative effects. Samples were collected at an interval of 4 h. Lake and stream water samples were collected weekly via grab sampling. A similar interval was used for groundwater, which was accessed with a submersible pump (COMET-Pumpen Systemtechnik GmbH & Co. KG, Pfaffschwende, Germany). For a period (16 September to 14 November 2020), the artificial fish ponds were also sampled as a potential water source for RWU. All liquid samples were extracted with a canula equipped syringe, filtered with a cellulose acetate filter (0.2 µm pores) into glass vials, stored in a fridge until analysis, and analysed via cavity ring-down spectroscopy (CRDS; Picarro L2130-i, Picarro Inc., Santa Clara, CA, USA) at the IGB laboratory.

Destructive sampling of bulk soil water was conducted monthly from June (after the first COVID-19 lockdown) to October 2020. Samples were collected at two locations: an open space with grass cover and a second location close to Northern Willow. Replicate samples were collected with a hand auger (diameter: 2 cm) at 0 to 10, 10 to 20, 20 to 40, 40 to 70, and 70 to 100 cm depth. Samples were filled in metalised bags and analysed using the direct equilibrium method from Wassenaar et al. (2008). Details of sample preparation can be found in Kleine et al. (2020). The samples were equilibrated for roughly 48 h before analysis. For correction, nine 10 mL standard water samples of $\delta^{18}$O ($-10.3$ ‰, $-7.68$ ‰, 2.91 ‰ or 1.53 ‰) and $\delta^2$H ($-72.81$ ‰, $-56.70$ ‰, 0.78 ‰ or 16.74 ‰) were used during every measuring routine. Soil water vapour analysis from the bags was conducted with a Los Gatos off-axis integrated cavity output spectroscopy (OA-ICOS) triple water-vapour isotope analyser (TWIA-45-EP, Los Gatos Research, Inc., San Jose, CA, USA). In this paper, we will refer to these samples measured with the direct equilibrium method as "bulk soil water".

Further, destructive sampling of xylem water was also conducted monthly from July to October 2020, together with the bulk soil water sampling. Each time, three sun-exposed branches were collected on Northern and Southern Willow, respectively. Only twigs with intact bark were collected and had their bark and phloem removed to prevent inter-

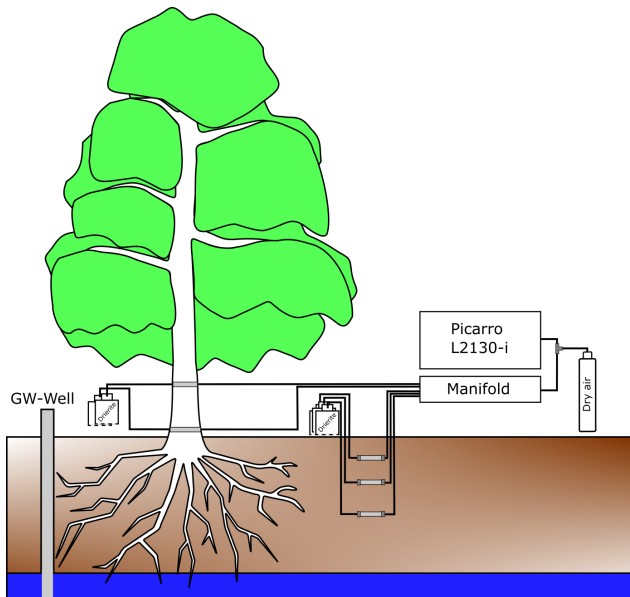

**Figure 2.** Conceptual graphic of the general in situ isotope measuring setup.

ference. Next, the twigs were cut inside 12 mL vials from Exetainer© and immediately sealed with a septum cap. In the field, the vials were stored in a polystyrene container and frozen after reaching the lab until cryogenic vacuum extraction. The cryogenic vacuum extraction was done according to Koeniger et al. (2011) but with 60 to 90 min of extraction time per sample. Before the extraction an empty vial is frozen with liquid nitrogen and evacuated by a vacuum pump. Next, the sample is frozen and connected to the evacuated vial via a stainless-steel capillary tube (Koeniger et al., 2011). Afterwards, to extract the water, the sample vial was placed in a 105 °C heated custom-built isolated aluminium block (Gaj et al., 2016). After extraction, the xylem water samples were measured using CRDS (Picarro L2130-i). We will further refer to the xylem water extracted by cryogenic vacuum extraction as "cryo-xylem water".

Stable water isotopes in soil water vapour were monitored via CRDS (Picarro L2130-i) in the two soil pits (Pit A and Pit B) at the three depths (Fig. 2). For vapour extraction in the soil, 7 cm long polypropylene membranes (0.2 µm pores) (Kübert et al., 2020) were used. In this paper, we will refer to the samples of the in situ soil pits as "in situ soil water".

Xylem water vapour was monitored with the same vapour sampling device from the two willows (Northern and Southern Willow) inside two stem boreholes each (Fig. 2). Xylem sampling was also delayed due to restrictions resulting from COVID-19 lockdowns. Consequently, measurements of Southern Willow started on 22 June 2020 and Northern Willow on 10 July 2020. At the beginning of the monitoring, both willows showed wounding effects, resulting in unrealistically high isotopic compositions declining exponentially over time. Hence, these time periods were excluded from further analysis. The boreholes were at 30 cm (further referred to as "lower") and 170 cm (referred to as "upper") height above the soil surface. As in the soil, polypropylene membranes were inserted into the boreholes, passing the stem horizontally (stem borehole equilibration). We will refer to all xylem water samples measured by this method as "in situ xylem water". The method of stem borehole equilibration was only tested before on tree logs and in a greenhouse experiment (Marshall et al., 2020) and a tropical forest in stems and roots of tropical trees (Kühnhammer et al., 2022). Therefore, the borehole equilibration method resembles a new design of in situ xylem water vapour. In the description by Marshall et al. (2020), the carrier gas was provided from one end, while an extraction tube of an isotope ratio infrared spectrometer (IRIS) for suction was set on the other side of the borehole, allowing for linear flow and equilibrium conditions (Marshall et al., 2020). Volkmann et al. (2016a) already used microporous polypropylene membranes. Their probe was embedded inside the sapwood and sealed for the inner and outer end of the borehole to prevent contamination with air water vapour (Volkmann et al., 2016a). Inside a mixing chamber of the probe, a gas–water mixture was sucked out by an extraction tube into an IRIS.

However, our new integration of the membrane approach from Volkmann et al. (2016a) into the stem borehole equilibration method of Marshall et al. (2020), which itself is novel, has the advantages of using dried ambient air as input flow for our boreholes with a reduced risk of fungal or bacterial infections of the tree due to the very small (0.2 µm) pores. Further, less artificial air is required that usually would have been needed to be carried into the field, hence making this setup easier compared to those needing constant gas supply from a bottle. Finally, using the same probe type (membrane inserted inside a matrix) makes the comparability of soil vs. xylem data easier. The boreholes had an inner diameter of 8 to 10 mm. Inside the membranes, fine PFA-sealed resistance thermometers (HSRTD, Omega Engineering, Norwalk, USA; tolerance: ±0.15 to 0.35 °C (over 0 to 100 °C)) were installed to measure the borehole temperature. The thermometers logged with the CR800 data logger and multiplexer. Daily mean temperature of the boreholes (required to convert vapour measurements into liquid isotopic composition) was similar to daily mean air temperature.

All soil and plant borehole tubes were attached to a bottle filled with desiccant (Drierite from W. A. Hammond DRIERITE Co. LTD, Xenia, OH, USA) to dry incoming air at one end and attached to the laser spectrometer (Picarro L2130-i) at the other end of the membrane. The boreholes were further sealed with waterproof glue (ORCA, Aquarium Münster Pahlsmeier GmbH, Telgte, Germany). In situ soil and xylem water was sampled successively for each point, and each sample was measured for 10 min in total at ~ 2 h intervals.

To avoid tube condensation, heating cables (ILLw.CT/Qx, Quintex GmbH, Lauda-Königshofen, Germany) were installed and wrapped with the tube in aluminium foil for insulation. The cables were controlled via an automatic multi socket (Gembird EG-PMS2, Gembird Software Ltd., Almere, the Netherlands) to prevent overheating in summer. To minimise condensation effects, the measurements were checked daily, and the system (manifold with attached tubes and membranes) was flushed for 10 min per probe to remove any water (Beyer et al., 2020). In the cases when condensation inside the system was identified, the respective data were discarded. We only used daily mean data to exclude sub-daily variance and produce a more reliable data set because the focus of our study is on seasonal variability of the willow's behaviour and environment. Daily means were calculated after all corrections and discarding of potential condensation affected data were complete.

## 3.4 Calibration and data analysis

Calibration of the in situ soil and xylem water system was achieved by a standard delivery module (Picarro A0101 Standards Delivery Module, Picarro, Inc., Santa Clara, CA, USA) using standards of known isotopic composition in $\delta^{18}O$ ($-10.41$‰, $-7.66$‰ or $1.45$‰) and $\delta^{2}H$ ($72.83$‰, $-55.86$‰ or $16.74$‰), respectively. These lab standards and all values of isotopic composition in this study are relative to Vienna Standard Mean Ocean Water (VSMOW). To correct for isotopic offsets and vapour concentration dependency, we used a similar approach to Schmidt et al. (2010) applying a linear regression of vapour concentration dependency slopes for $\delta^{2}H$ and $\delta^{18}O$ and of the slopes on the $\delta$ values for $\delta^{2}H$ and $\delta^{18}O$ isotopic offset correction. We only used measured water vapour concentrations and added linear regressions of temperature dependency slopes. However, the bulk soil sample results suggested that stronger water concentration dependencies were not fully corrected with this approach; hence, we applied polynomial regressions on measured isotopic compositions for different temperatures. Thus, we were able to correct highly variable water concentrations by using the soil pit temperatures of the specific depths.

To check for $CO_2$ contamination, we used the linewidth variable ("h2o_vy") of the raw data from the Picarro instrument as suggested by Gralher et al. (2016). We found a strong correlation between this variable and the water concentration measured by the laser spectrometer, which indicated that no significant $CO_2$ contamination was detected.

Furthermore, we assumed the liquid water source to be in equilibrium with the measured water vapour, allowing us to calculate the liquid–vapour fractionation by a model of type 1 (Majoube, 1971):

$$\alpha = \exp \frac{a\left(\frac{10^6}{T_k^2}\right) + b\left(\frac{10^3}{T_k}\right) + c}{1000}, \tag{2}$$

where $\alpha$ is the isotopic fractionation factor, $T_k$ is the temperature (in K), and $a$, $b$, and $c$ are empirical parameters that vary depending on the isotopologue.

The deviation of $\delta^{2}H$ and $\delta^{18}O$ composition in the sample from the global precipitation (or global meteoric water line, GMWL) is presented as deuterium excess (short d-excess) by Dansgaard (1964):

$$\text{d-excess} = \delta^{2}H - 8 \cdot \delta^{18}O. \tag{3}$$

The d-excess indicates fractionation processes of the sample compared to the global precipitation. To investigate the local evaporative effects of the samples water, isotopologues were used to calculate the line-conditioned excess (short lc-excess) (see Landwehr and Coplen, 2006). The lc-excess describes the deviation of the sample from the local meteoric water line (LMWL):

$$\text{lc-excess} = \delta^{2}H - a \cdot \delta^{18}O - b, \tag{4}$$

where $a$ is the slope and $b$ the intercept of the weighted isotopic composition of the local precipitation (IGB: $a = 7.4$, $b = 3.7$). The LMWL was calculated by linear regression of the precipitation stable water isotopes (plotting $\delta^{18}O$ against $\delta^{2}H$) measured at IGB from May 2020 to January 2021 ($R^2 = 0.98$). The similarity of soil water isotopic compositions measured with in situ and destructive sampling was assessed by calculating the Euclidean distance to the 1 : 1 line of $\delta^{18}O$ in situ vs. $\delta^{18}O$ direct equilibrium ("bulk") or cryogenic extracted and $\delta^{2}H$ in situ vs. $\delta^{2}H$ direct equilibrium or cryogenic extracted for soil and xylem signatures, respectively.

## 3.5 Mixing model

To quantify sources of in situ xylem water from soil water of different depths, we used the Bayesian isotopic mixing model, SIAR (Parnell et al., 2010). The download of the mixing model SIAR is available in the packages section of the Comprehensive R Archive Network site (CRAN) – https://cran.r-project.org/web/packages/siar/index.html (last access: 1 April 2022). Following an initial assessment of potential water sources (see below), the soil isotopes from 10, 40, and 100 cm from Pits A and B were used. These soil depths likely represent the majority of the root distribution of willows due to high near-surface rooting densities (Cunniff et al., 2015; Marttila et al., 2017). To establish if the source changes through the growing season, soil and xylem isotopes were divided into weekly bins beginning 21 June 2020, and SIAR was run independently for each weekly group of sources and in situ xylem water. To constrain the mixing model, $\delta^{2}H$, $\delta^{18}O$, and d-excess (d-excess $= \delta^{2}H - 8 \cdot \delta^{18}O$) were simultaneously used for the soil and vegetation. To optimise the soil source water to in situ xylem water, Markov chain Monte Carlo approaches were utilised with 500 000 simulations, using the first 50 000 as burn-in results (which

were then discarded). No prior information was provided to the model (uniform a priori distribution) for each soil depth. For Southern Willow, soil isotopes for each depth were averaged for Pit A and B as the relative similarities between Pit A and B would lead to limited identification of soil source between sites. To evaluate the efficiency of the SIAR model for identifying the proportions of source waters, the mean absolute error (MAE) was evaluated for each weekly bin using proportion-weighted soil isotopes and vegetation isotopes. Proportion-weighted soil isotopes were estimated as

$$\delta_{mix} = P1 \cdot \delta_{10\,cm} + P2 \cdot \delta_{40\,cm} + P3 \cdot \delta_{100\,cm}, \tag{5}$$

where $P1$, $P2$, and $P3$ are the proportion of water from 10, 40, and 100 cm, respectively, as estimated by SIAR.

## 4 Results

### 4.1 Hydroclimatic conditions

The study period was characterised by 11 precipitation events with $> 10\,mm\,d^{-1}$ (Fig. 3a). The largest event was in late August, while the longest dry period occurred in September. Daily mean air temperatures ($T$) gradually increased until mid-August and decreased afterwards. Vapour pressure deficit (VPD) as a driver of transpiration correlated with $T$ but also depended on water vapour pressure (high $T$ and low water vapour pressures cause high VPD). Maxima occurred in the beginning of June and mid-August; in late August the VPD declines alongside $T$. Evapotranspiration (ET) had minima in times of low precipitation amount and high $T$, and this was reflected in sap flow rates (Fig. 4a). ET increased after large precipitation events when moisture availability increased. At the end of the measuring period, as $T$ and VPD decreased, ET decreased as well.

The volumetric water content (VWC) of the soils at 10 cm depths strongly responded to precipitation events (Fig. 3e and f). Despite the different time series, it became clear that this variability was more distinct in Pit B (min: 5.7 %, max: 21.0 %, Fig. 3e) than in A (min: 4.0 %, max: 11.4 %, Fig. 3f). Generally, Pit A (beneath the tree canopy) was much drier than B. A precipitation event on 13 June resulted in a significant increase of soil moisture in Pit A, with rapid increases in VWC even at 40 cm. VWC responses to precipitation at 100 cm depth were very damped, with little change detected in both pits (Pit A min: 12.6 %, max: 14.2 %; Pit B min: 17.6 %, max: 20.7 %), even after the August event.

### 4.2 Vegetation growth dynamics and transpiration

Daily sap flow ranged from 5000 to $71\,000\,cm^3\,d^{-1}$ (5 to $71\,L\,d^{-1}$) for Northern Willow and 4000 to $86\,000\,cm^3\,d^{-1}$ (4 to $86\,L\,d^{-1}$) for Southern Willow (Fig. 4a). This compares well, in terms of order of magnitude, to other willows shown to vary between 30 and $100\,L\,d^{-1}$ during the grow-

ing season (Schaeffer et al., 2000). Sap flux in Northern Willow increased from the beginning of the measuring period until 30 June, after which it dropped. Sap flux in Southern Willow increased until 13 August and then decreased until 4 September, followed by an increase until 24 September. High ET values (Fig. 4a) were generally reflected in higher sap flow rates, though ET (recorded below the tree canopy) was low at a time of high sap flow in August, reflecting low ET from grassland soils. Both trees grew consistently with daily stem radius incrementation following leaf-out until the end of the growing season at $\sim$ 15 September (Fig. 4b). Sub-daily shrinking and swelling of the stem occurred, though this does not show in the averaged daily data set presented in this study. The main growing phase started around June, and total growth during the measuring period was 12.8 mm for Northern Willow and 16.4 mm for Southern Willow.

### 4.3 Dynamics in stable water isotopes

The daily average of the in situ isotope values showed strong links between larger ($> 10\,mm\,d^{-1}$) precipitation events and soil water $\delta^2H$ data (Fig. 5a and b; $\delta^{18}O$ is shown in Fig. S1 in the Supplement). The highest fluctuations occurred in the upper soil layer, while the isotopic composition at 40 and 100 cm both showed very damped responses to precipitation. At 40 cm, responses were more marked in canopy-covered Pit A than Pit B (Fig. 5), probably caused by the relatively smaller water stored in the soil of Pit A compared to Pit B, due to the canopy cover and more interception at Pit A. Due to these generally smaller water amounts, incoming precipitation affected the isotopic composition of the topsoil water of Pit A more than at Pit B.

In contrast, the in situ xylem water composition did not show immediate response to precipitation events, though the shorter run of data makes it difficult to interpret longer-term variations, especially for the Northern Willow. However, the longer time series of the Southern Willow suggests that the xylem composition, like the soil at 10 cm, became more depleted after summer rain in July, before slowly increasing again through August and September (Fig. 5b). Comparing the in situ xylem water of the two probes on each tree (upper and lower), they were mostly very similar to one another.

In situ soil water lc-excess was most negative with again higher variability in canopy-covered Pit A than in Pit B at a soil depth of 10 cm (Fig. 6). In Pit A, lc-excess exhibited variability that was consistent with cycles of wetting and evaporative drying causing negative values (Fig. 6b). A similar, though highly damped trend was apparent at 40 cm, while at 100 cm the lc-excess remained close to zero, with a suggestion of more fractionated water in the latter part of the study from late September. Variations were more damped in Pit B and at 10 cm lc-excess could become positive following rainfall (Fig. 6c). By late summer, lc-excess at 40 and 100 cm was gradually becoming more negative, consistent with the

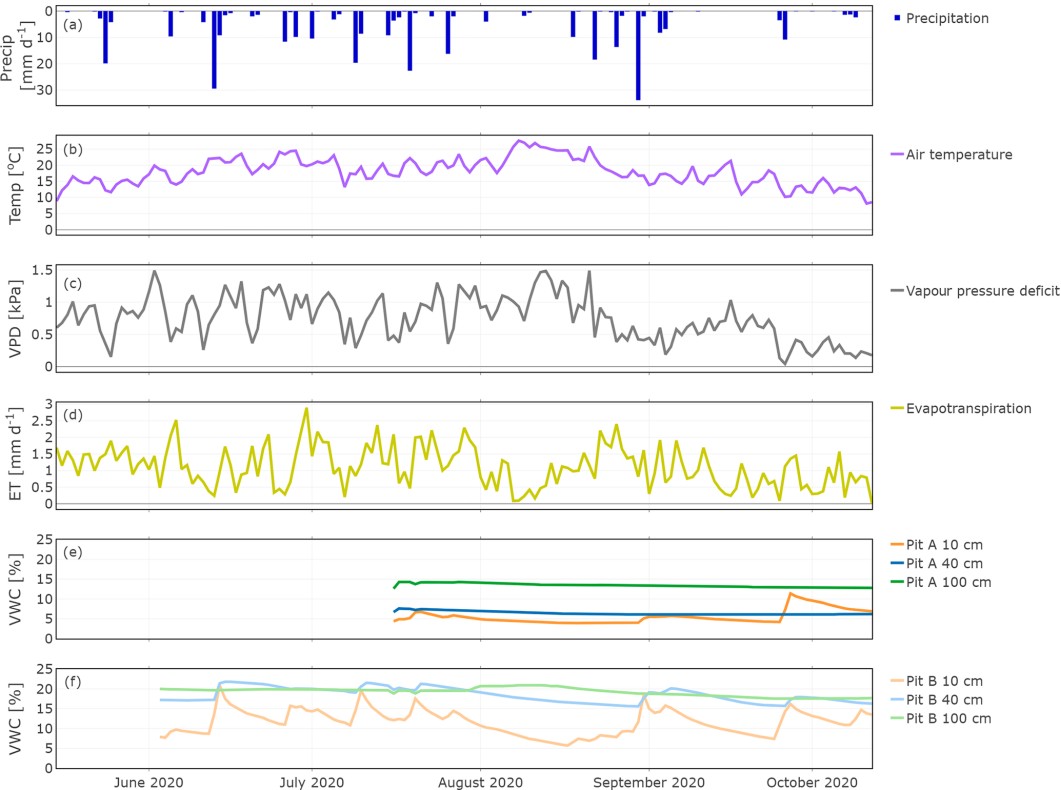

**Figure 3.** Hydroclimatic conditions showing daily precipitation **(a)**, air temperature **(b)**, vapour pressure deficit **(c)**, evapotranspiration **(d)**, and soil moisture for Pit A **(e)** and Pit B **(f)**.

percolation of more fractionated water from the upper soil with successive wetting fronts in response to rainfall inputs.

In situ xylem water lc-excess in Southern Willow varied around $-15$‰ until mid-August where the lc-excess increased constantly up to 0.2‰. In Northern Willow, in situ xylem water lc-excess became enriched from the end of August to October.

In the dual isotope plot (Fig. 7), all water compartments, except precipitation, mostly lie below the LMWL (local meteoric water line). The largest variability was found in precipitation, while the variability of groundwater and stream water was restricted. This plot also nicely illustrates the substantial overlap between the isotopic compositions of xylem and near-surface soil water measured in situ.

The range of stable isotope values for all measured water compartments (precipitation, soil water, groundwater, and lake and stream water) is also summarised in Fig. 8 and Table 1. These show again the large variability of precipitation and the very narrow range of stream, lake and groundwater samples ($-53.2$‰ to $-42.8$‰ for $\delta^2$H and $-7.1$‰ to $-5.2$‰ for $\delta^{18}$O). The lc-excess (Fig. 8c) of precipitation varied roughly between $-10$‰ and 10‰, while mean values of stream and lake water were $\approx -7$‰, reflecting the influence of fractionation. Groundwater lc-excess was mostly slightly positive. The uppermost in situ soil water of both Pit

A and B was most enriched (up to $-10.4$‰ for $\delta^2$H and 0.0‰ for $\delta^{18}$O) and showed larger variabilities compared with deeper in situ soil waters. This was generally similar to the bulk soil water (Fig. S2 in the Supplement). The upper in situ soil water of the drier Pit A was more variable and enriched during the measuring period than in Pit B. At 100 cm depth, in situ soil water of both pits showed a very narrow range of isotopic compositions (Table 1), with water from Pit A being roughly 10‰ more enriched in $\delta^2$H ($\sim 1$‰ in $\delta^{18}$O) compared to Pit B. Like the bulk soil samples, the in situ results were more depleted in deeper layers (Table 1). The in situ soil water lc-excess of the top layer of the canopy-covered Pit A was in general most negative, while water from open-spaced Pit B at 10 cm even reached positive lc-excess values.

Detailed bulk soil water composition is shown in heat maps of $\delta^{18}$O, $\delta^2$H and lc-excess from the "canopy-covered" (NW of Northern Willow) and "open space" (south of Pit B) sample sites (Fig. S3 in the Supplement). Both locations showed variation with depth and in time in response to rainfall inputs and changing ET. Usually, the bulk water at 10 cm was most enriched in $\delta^{18}$O and $\delta^2$H, while at 40 to 70 cm it was most depleted. Canopy-covered bulk soil water was, in general, more enriched in $\delta^{18}$O than in Open Space, resulting in differences in lc-excess. The mean Euclidean distance

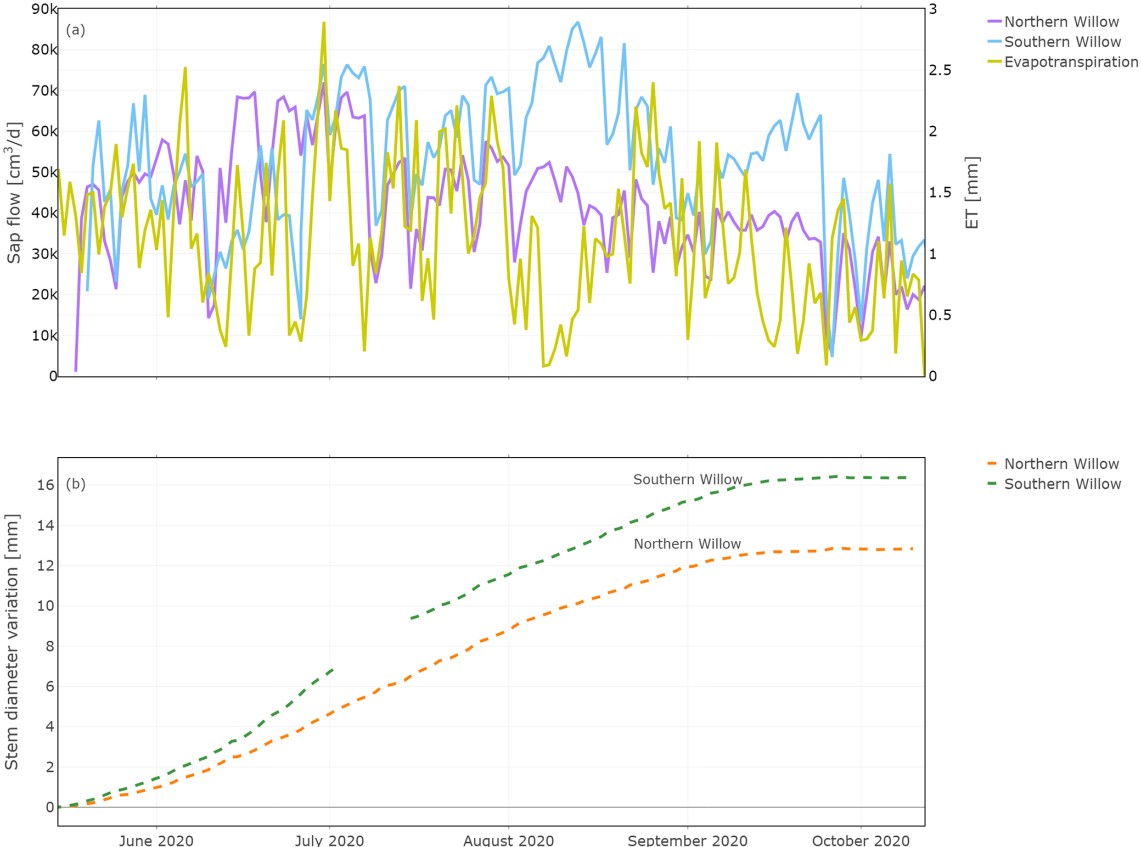

**Figure 4.** Daily total sap flow measured at the southern stem side and stem diameter variation measured for the stem radius at the northern stem side. Plot **(a)** shows the results of sap flow together with evapotranspiration for comparison and **(b)** the results of the stem diameter variation during the measuring period.

as a measure of similarity between bulk soil and in situ data showed $\sim 0.7\,‰$ for $\delta^{18}$O and $\sim 4.6\,‰$ for $\delta^2$H (see Table 2).

The in situ xylem water showed most similarities with the topsoil signatures (Figs. 7 and 8). Like the upper in situ soil water, the in situ xylem water varied over wider ranges (e.g. $-29.2\,‰$ to $-1.9\,‰$ for $\delta^2$H and $-5.4\,‰$ to $-0.9\,‰$ for $\delta^{18}$O of Northern Willow upper), with its distribution mostly being in the range of the signatures at canopy-covered Pit A at 10 cm. The in situ xylem water lc-excess values also showed a wide range ($-28\,‰$ to $0.2\,‰$); still the values mostly fit inside the total lc-excess range of in situ soil water from Pit A 10 cm. The xylem water isotopic composition showed diurnal fluctuations that were also affected by sap flow activity of the willows (high sap flow resulted in heavier isotopic compositions but also higher water concentrations in the measurement).

Cryogenically determined xylem water isotopic composition from the destructive sampling (Figs. S4 and S5 in the Supplement) was more depleted compared to the in situ xylem water (Table 1). The lc-excess of the cryo-xylem water was also more enriched compared to the in situ xylem water (Fig. S4). Like the in situ xylem water, the cryo-xylem water

isotopic composition was comparable to soil isotopic composition at 10 cm but indicated less evaporated conditions. After correcting the cryo-xylem water $\delta^2$H data by adding $8.1\,‰$ (Chen et al., 2020), the cryo-xylem $\delta^2$H results were more comparable to the in situ ones. However, after this correction the lc-excess values drifted further apart from the in situ xylem water (Fig. S5). The mean Euclidean distance as a measure of similarity between cryo-xylem and in situ xylem water showed $\sim 1.6\,‰$ for $\delta^{18}$O and $\sim 12.1\,‰$ for $\delta^2$H (or $\sim 6.8\,‰$ for $\delta^2$H after correction by Chen et al., 2020) (see Table 2).

### 4.4   Determining potential xylem water sources

The SIAR Bayesian mixing model estimated the possible sources of RWU (i.e. integrated in the composition of xylem) of the willows from isotope measurements at specific soil water depths (Fig. 9). The mixing model indicated that the xylem composition in both willow trees could be explained by uptake of water almost solely from the topsoil (10 cm) for most of the study period, regardless of soil moisture variations and with low uncertainty (as indicated by the standard deviations in the plot). This is consistent with the qualitative

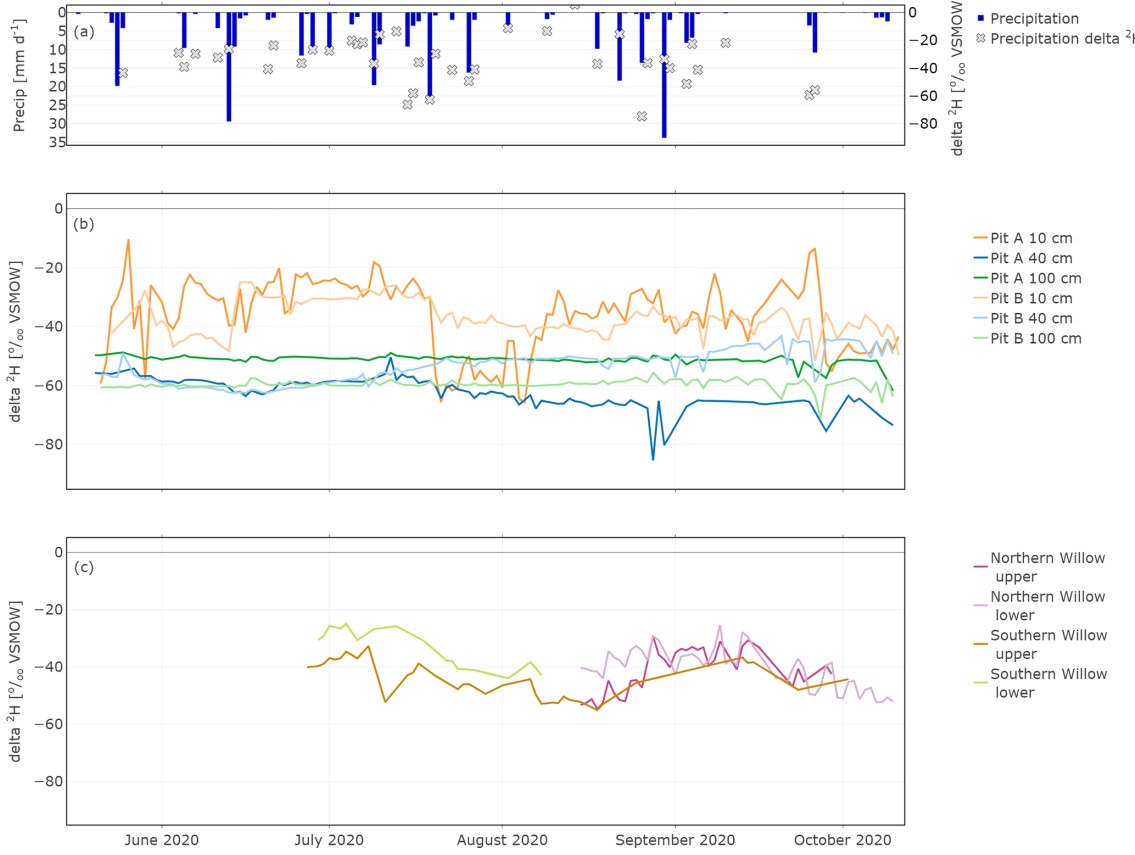

**Figure 5.** In situ time series of daily $\delta^2$H in precipitation (**a**), Pit A and Pit B (**b**), and Northern Willow and Southern Willow (**c**).

**Table 1.** The amount ($n$) and 5th, 50th (median), and 95th percentiles of $\delta^{18}$O and $\delta^2$H (‰ VSMOW) signatures. Results are presented for precipitation, groundwater, surface water (which is lake and stream water combined), in situ soil water at 10, 40, and 100 cm combined for both pits (named "Soil 10", "Soil 40", and "Soil 100"), bulk soil water at canopy-covered and open space at 10, 40, and 100 cm (named "Canopy-covered 10", "Canopy-covered 40", "Canopy-covered 100", "Open space 10", "Open space 40", and "Open space 100"), in situ xylem water at the upper and the lower borehole combined for both willows (named "Tree upper" and "Tree lower"), and cryogenic extracted xylem water combined for both willows (named "Tree cryo").

| Sample | | $\delta^2$H ‰ VSMOW | | | $\delta^{18}$O ‰ VSMOW | | |
|---|---|---|---|---|---|---|---|
| | $n$ | 5th percentile | Median | 95th percentile | 5th percentile | Median | 95th percentile |
| Precipitation | 40 | −62.73 | −34.67 | −13.24 | −9.03 | −4.79 | −1.70 |
| Groundwater | 23 | −52.87 | −49.73 | −45.36 | −6.99 | −6.35 | −5.57 |
| Surface water | 40 | −48.75 | −45.34 | −43.90 | −6.18 | −5.62 | −5.24 |
| Soil 10 | 265 | −56.08 | −36.44 | −24.13 | −7.14 | −4.72 | −1.53 |
| Soil 40 | 242 | −66.48 | −58.52 | −45.36 | −8.64 | −7.82 | −6.12 |
| Soil 100 | 261 | −60.49 | −57.60 | −50.21 | −8.12 | −7.53 | −6.89 |
| Canopy-covered 10 | 6 | −53.76 | −35.93 | −32.37 | −7.97 | −4.65 | −2.05 |
| Canopy-covered 40 | 12 | −62.04 | −53.30 | −43.56 | −8.67 | −7.30 | −5.56 |
| Canopy-covered 100 | 12 | −65.44 | −61.82 | −59.03 | −9.04 | −8.42 | −7.95 |
| Open space 10 | 6 | −55.08 | −40.37 | −33.32 | −7.40 | −5.08 | −4.53 |
| Open space 40 | 12 | −61.91 | −52.95 | −42.51 | −8.05 | −6.98 | −5.63 |
| Open space 100 | 12 | −65.39 | −63.52 | −59.86 | −8.65 | −8.34 | −7.74 |
| Tree upper | 73 | −53.01 | −42.86 | −32.38 | −6.15 | −3.94 | −2.28 |
| Tree lower | 73 | −51.08 | −38.51 | −25.96 | −7.01 | −3.72 | −1.76 |
| Tree cryo | 29 | −63.07 | −55.63 | −46.34 | −7.61 | −5.85 | −4.68 |

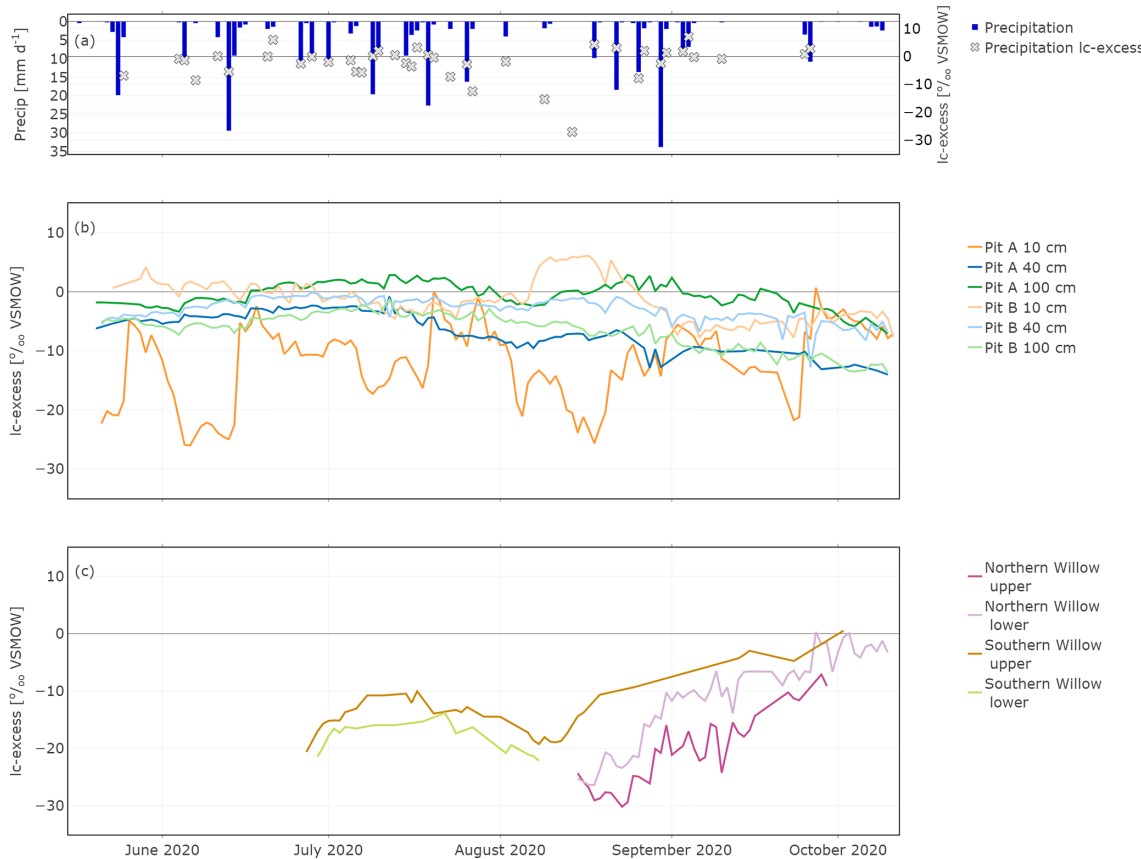

**Figure 6.** In situ time series of daily lc-excess in precipitation **(a)**, Pit A and Pit B **(b)**, and Northern Willow and Southern Willow **(c)**.

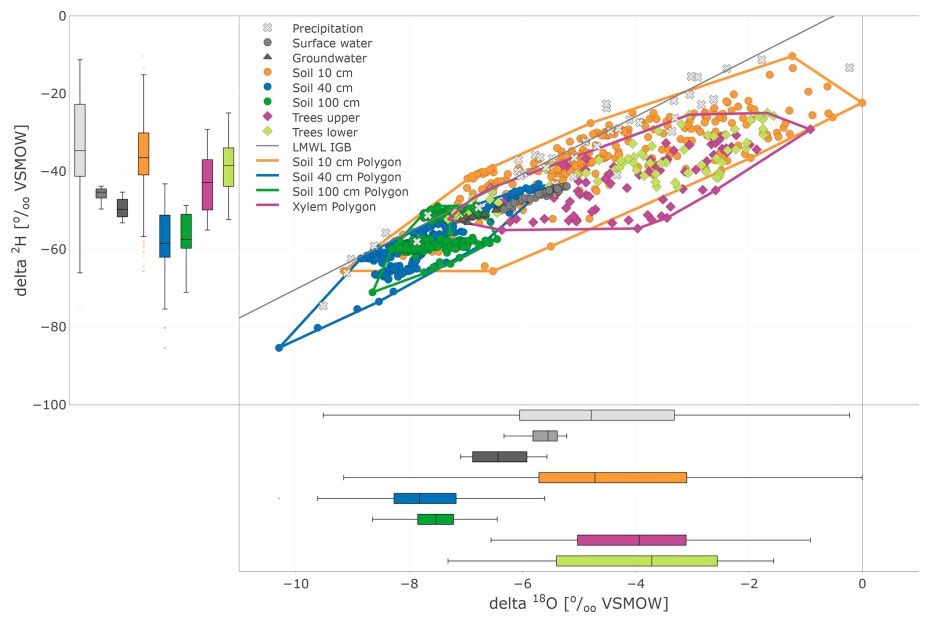

**Figure 7.** Dual isotope plot of in situ (daily) soil and xylem as well as precipitation (daily), surface water, and groundwater (weekly) sampling. Soil and tree data are highlighted with boundary polygons for 10, 40, and 100 cm and tree (upper and lower results joined) clusters. Additional box plots show the sample distribution of the data sets.

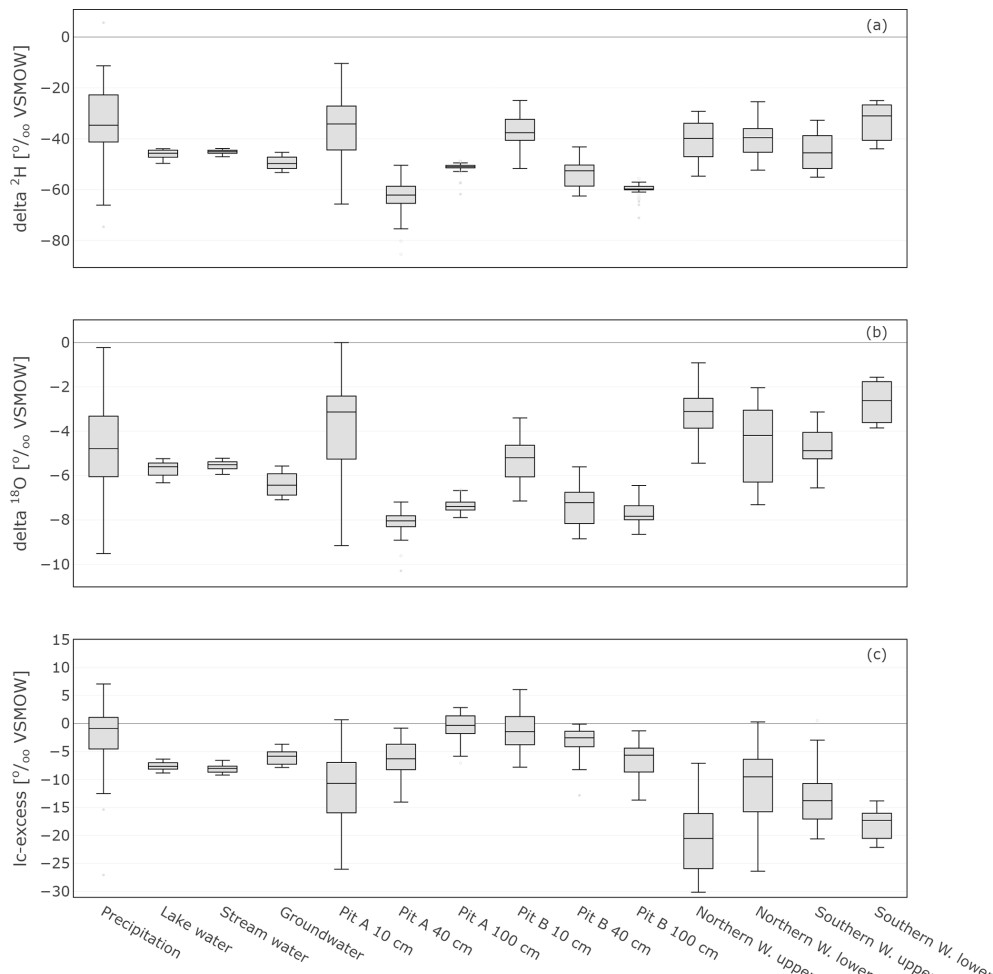

**Figure 8.** Box plots showing the isotopic composition as well as lc-excess of daily precipitation sampling, weekly sampled groundwater, lake and stream water, in situ sampled soil, and xylem water. The lc-excess is an indicator of evaporational effects and describes the offset of the sample from the LMWL (see Sect. 3.4).

**Table 2.** Variance of Euclidean distance between in situ and direct equilibrium ("bulk") soil stable water isotopes and between cryogenic extracted twig and in situ xylem stable water isotopes. Euclidean distances were calculated between the $\delta^2$H (or $\delta^{18}$O) bulk (or cryogenic extracted) and $\delta^2$H (or $\delta^{18}$O) in situ point and the 1 : 1 line. In situ data were taken from the day of sampling bulk (or twig) samples. The results from "cryo cor" refer to corrected $\delta^2$H values where 8.1‰ was added (see Chen et al., 2020).

|  |  | $\delta^2$H ‰ VSMOW | | | $\delta^{18}$O ‰ VSMOW | | |
|---|---|---|---|---|---|---|---|
|  | $n$ | Min | Mean | Max | Min | Mean | Max |
| Bulk vs. in situ | 34 | 0.01 | 4.62 | 12.12 | 0.00 | 0.68 | 2.00 |
| Cryo vs. in situ | 7 | 5.13 | 12.08 | 18.01 | 0.07 | 1.62 | 3.24 |
| Cryo cor vs. in situ | 7 | 1.20 | 6.81 | 12.89 | 0.07 | 1.62 | 3.24 |

overlap of upper soil water and in situ xylem water in the dual isotope space in Fig. 7. However, comparing the cumulative tree transpiration and rainfall to soil water storage, it would have also been possible for the willows to solely take water up from the top 40 cm. Through the soil and xylem water signatures, the model was able to predict RWU as mainly water from the top 10 cm of the soil. RWU from the topsoil by the willows probably led to the observed dry conditions in the topsoil, even though ET was not very high (max. 2.9 mm), and the site was often shaded. At the end of September and in October, there was a suggestion that deeper water sources might have become more important, though results became

increasingly uncertain. By this time, only 60 % to 95 % could be accounted for by uptake from the topsoil, while 0 % to 10 % and 0 % to 40 % were taken from 40 and 100 cm depth, respectively, in Northern Willow. In Southern Willow, the inferred increased uptake of soil water from 40 and 100 cm occurred 1 week later, with $\sim 70\,\%$, $\sim 25\,\%$ and $\sim 5\,\%$ taken from 10, 40, and 100 cm, respectively. Overall, mean absolute errors for the prediction of in situ xylem water composition by the SIAR model were small, being 6.5 ‰ and 8.8 ‰ for $\delta^2$H and 1.8 ‰ and 1.3 ‰ for $\delta^{18}$O for Southern Willow and Northern Willow, respectively (Fig. S6 in the Supplement).

## 5 Discussion

### 5.1 Dynamics of sources of root water uptake

Both qualitative assessment and quantitative analysis of the isotope data with the Bayesian mixing model infer that RWU by the willows was predominantly sourced from the upper soil horizons during the period when xylem water was analysed in situ. This water tended to be more enriched in heavier isotopes, showing the effects of evaporative fractionation. There was nothing to indicate that deeper, more depleted groundwater or stream water sources were major components of RWU during the study period. There was some evidence from the mixing analysis that trees used deeper soil water to supplement shallow sources later in the monitoring period, which coincided with a drier spell in September, though soil moisture availability in the upper soil profile of the pits did not differ significantly compared to hotter summer months like July or August. Although replication was limited, the two soil pits showed marked spatial differences in both soil moisture content and the variability in soil water isotopes, though similar changes with depth were apparent. This likely reflects the effects of interception losses below the tree canopy greatly reducing the moisture content of Pit A. This low storage volume, in turn, resulted in more marked effects of both evaporative fractionation and mixing of infiltrating rainfall on the soil water isotope signals. Interestingly, individual rainfall events had no significant effects on xylem water isotopic composition.

These findings are consistent with other studies that have shown that willows mainly utilise water from the near-surface soil horizons (Marttila et al., 2017), which is where nutrient availability is highest (Goldsmith et al., 2012), and most tree species adapt their greatest fine root densities to water and nutrient supply (Hertel et al., 2013). Similarly, Nehemy et al. (2021) found that willows use shallower water sources until drier conditions force water uptake from deeper soil layers. In the current study, it is not totally clear why the results of the mixing model in September showed a shift from topsoil to deeper soil layers, even though the soil moisture of the topsoil started to rewet again. This shift comes at the end

of the growing season, shortly after stem incremental growth ceases, and may be indicative of physiological changes at the onset of autumn. As was the case for beeches in Gessler et al. (2022) it is possible that the total amount of water taken up from 10 cm soil was reduced at the end of the growing season, shifting the relative contributions of deeper soil waters. The decline in sap flow in October (Fig. 4a) also fits such a scenario. However, it should be noted that the summer of 2020 was not characterised by a prolonged period of water stress, such as the well-documented drought summer of 2018 (Kleine et al., 2020; Gessler et al., 2022). As shown by their constant growth, high sap flow, and relatively low ET, the willows did not experience water stress during our field experiment. Under drought conditions, a very different water uptake strategy of the willows might be evident, underlining the need for multi-year ecohydrology studies that capture hydroclimatic variability. However, other studies have shown that soils and not groundwater are the most likely sources of RWU, even if the latter might be expected (Brooks et al., 2010; Evaristo et al., 2015). Furthermore, recent isotopic mixing studies (e.g. Gessler et al., 2022) showed that some tree species (beech) are not able to compensate for drought-reduced topsoil water availability by taking up more water from deeper soil layers, emphasising the importance of investigating different species individually. A labelling field experiment by Kühnhammer et al. (2022) showed a general response of xylem water isotopic composition to irrigation events but no direct response to single events, which is similar to our findings.

It should be stressed that the mixing model used here is relatively simplistic and coarse as it does not fully characterise the potential heterogeneity in soil properties and root distribution (see Sprenger and Allen, 2020). For example, one of the conditions for the Bayesian mixing model assumes all endmembers are known. Since we conducted a field experiment under natural conditions, we cannot exclude unknown water sources completely. Even though we have covered obvious sources of precipitation, local soil water, groundwater, and lake and stream water, roots may be able to access other sources. For example, modelling work with the data used in this study showed that the willows likely have a large horizontal range of fine roots ($> 6$ m) and may access more distant heterogenous water sources (Smith et al., 2021). Furthermore, numerous recent studies have shown that stem xylem water likely reflects the integrated effects of uptake over many weeks or months, depending on the species, age/size, seasonality, and antecedent hydroclimatic variability (e.g. McCutcheon et al., 2017; Tetzlaff et al., 2021; Snelgrove et al., 2021). Moreover, recent modelling work has shown that storage, mixing, and remobilisation of RWU in trees may explain a lack of direct correlation between soil water and xylem water (Knighton et al., 2020). Therefore, it is possible that the increase of deeper soil water in the stem water pool in September was observed weeks after its root uptake. Condition-controlled experiments with labelled water could

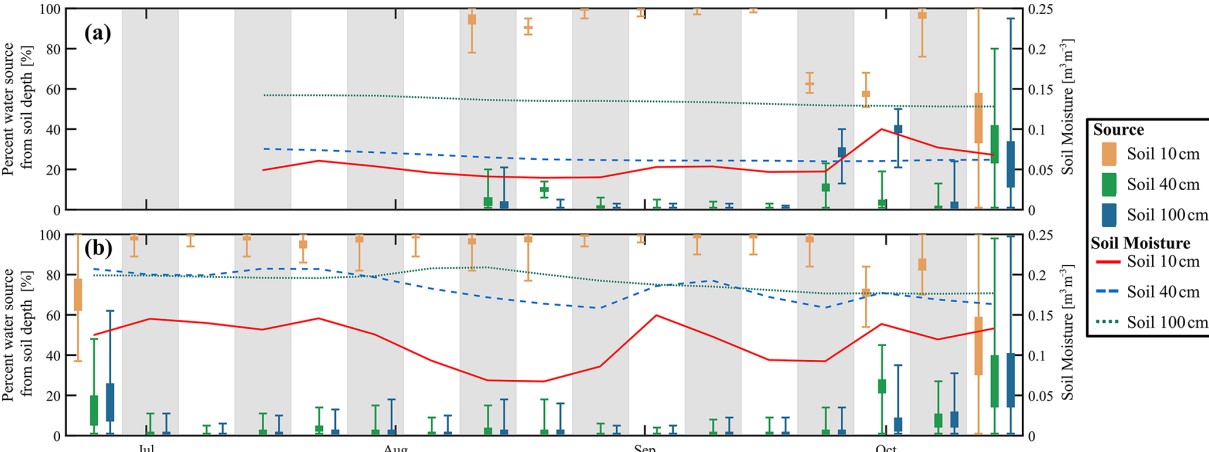

**Figure 9.** Estimated percentage of willow water from each soil depth in **(a)** Northern Willow and **(b)** Southern Willow. The box plots for each source show the 25th and 75th percentiles, with whiskers extending to the minimum and maximum estimated proportion. Grey and white bands show the division of each week. The right-hand *y* axis shows the soil moisture for each soil depth (weekly average).

be helpful to further investigate such possible temporal integrations of water signatures. However, the mixing model used in this study still provides a reasonable first approximation of the willows studied here. It possibly helps that willows have a rapid metabolism, supported by the high sap flow velocities and steady, sustained growth over the study period. Moreover, the two young, same-aged trees on an immature freely draining soil provided a relatively simple system with more limited variability in the xylem isotope signal between and within the trees than might be expected in a more heterogenous and natural riparian forest. More complex ecohydrological modelling that explicitly conceptualises the travel time of water in trees may be able to provide deeper insights (e.g. Mennekes et al., 2021). Finally, moving window analysis connecting a time window of the soil water to the best fitting time window of the xylem water might also be an easy but useful approach to evaluate the temporal variability.

## 5.2 Using in situ monitoring over extended periods: potential and challenges

In situ monitoring is usually characterised by relatively short-term periods (days to weeks) monitoring campaigns, often under controlled experimental conditions (Volkmann et al., 2016a; Gaj et al., 2016; Kübert et al., 2020). More recent work has demonstrated the feasibility of field application of spectroscopes in direct in situ monitoring to produce longer-term, reliable data of water in different compartments of the critical zone (Mennekes et al., 2021; Seeger and Weiler, 2021; Kühnhammer et al., 2022). Our study contributes to this evolution of such isotopic techniques to directly observe ecohydrological processes in soil and vegetation systems. In terms of soil isotope monitoring, our approach was very similar to that of Oerter and Bowen (2017), though we used ambient atmosphere as carrier gas rather than $N_2$. Consequently,

we needed to dry the air using desiccant, and similar to Volkmann et al. (2016a), this allowed us to achieve 2-hourly sampling with stable plateaus of isotope ratios from each inlet in the system (Fig. 2). For seasonal investigations, we suggest that fewer sub-daily measurements than we took (but at least two or three times a day) may be sufficient to capture diurnal variation. It might be useful to link the timing of these measurements to sap flow activity of a tree (e.g. one measurement at maximum and minimum daily sap flow, respectively). Of course, such an approach needs further investigation, especially for different species.

For the xylem water sampling in the trees, we adapted the method proposed by Marshall et al. (2020), with minor changes being the first usage in angiosperms of a temperate climate (see Sect. 3.3). The high-frequency results present an improvement on results gained by destructive sampling, where questions remain over exactly what is extracted by cryogenic methods (Chen et al., 2020; Barbeta et al., 2020). Our results showed moderate to no comparability between in situ and cryogenic extracted measurements. Similar results in cryogenic vs. in situ extracted results were found elsewhere (Gessler et al., 2022; Kühnhammer et al., 2022), but in others, findings differed (Mennekes et al., 2021). Possible reasons for differences in in situ vs. cryogenic extracted xylem water isotopic composition can be released cell content due to the defrosting process, organic contamination (if measuring with an IRIS), and heterogeneity of the tree trunk. In terms of the in situ measurements, a much higher variability in the data or slight, undetected condensation effects, or non-equilibrium conditions might also result in offsets to cryogenic extracted results. Here, we observed no offset in $\delta^{18}O$ (see Volkmann et al., 2016a; Marshall et al., 2020; Gessler et al., 2022), and tests of a line-width-related variable (as suggested by Gralher et al., 2016) showed no effects of organic or $CO_2$ contamination (see Beyer et al., 2020). However, we

did see individual differences of the willows showing the same trend of xylem water isotopic composition, while the absolute values differed. For a prolonged installation to ensure equilibrium conditions, it seemed advantageous to locate the boreholes further up the tree stem, where the increment of radial growth is lower as the stem narrows.

A particular challenge for in situ monitoring during the growing season is cooling and keeping the instrument $< 35\,^{\circ}\mathrm{C}$ in high summer temperatures. This was achieved using a sunshade over the equipment and computer cooling fans. Despite this, the isotope time series for in situ water vapour measured in both soils and xylem showed diurnal variation, which, although reflecting natural processes, did include sampling artefacts that needed to be quality controlled. In the current paper we have not focused on diurnal variations, as the seasonal changes in RWU have been the main focus, and data were averaged to daily time steps. Nevertheless, diurnal cycles of temperature and humidity not only contributed to diurnal variation in measurements, but also caused condensation in tubing (e.g. Beyer et al., 2020; Kühnhammer et al., 2022) that could not always be prevented despite heating the tubes (see above). As a result, some data points were rejected, but still sufficient data were collected to estimate daily means. This was a particular problem in cooler nights (see Gaj et al., 2016) and following rainfall, especially in late summer and autumn as air temperature drops faster compared to soil temperature, causing a temperature gradient. The effects were most marked in the topsoil layer and xylem where the sampling tubes experienced the most marked temperature variations. For much of the monitoring period, flushing the system (manifold, tubes and probes) with dry air each morning was needed. However, this daily maintenance work is of course highly labour-intensive.

In this study we focused on monitoring natural abundance of isotopes in various critical zone compartments to better understand interactions in the soil–plant–atmosphere continuum of a plot of riparian willow trees. Other studies have reported tracer experiments that focused on specific hypothesis tested under controlled conditions (e.g. Seeger and Weiler, 2021) or field experiments with natural conditions but additional labelling (e.g. Kühnhammer et al., 2022). We see these approaches as being complementary, and our results underline the usefulness of investigating natural abundances of isotopes in critical zone compartments, especially if combined with auxiliary ecohydrological measurements. Although clear tracer "breakthrough" signals are much more evident in labelling studies, there are advantages to observing conditions as they occur in nature and the realistic reaction of trees to its new rainwater inputs in relation to the distribution of the water sources.

## 6 Conclusions

We conducted an in situ field study of stable water isotopes in soil water and the xylem of willow trees in conjunction with hydroclimatic monitoring and measurements of sap flow, stem size, soil temperature and moisture, and stable water isotopes in precipitation, lake water, stream water, and groundwater. Our investigation delivered reliable high-frequency stable water isotope data in two soil pits at three depths and two willows in two stem boreholes with a $\sim$ 2-hourly resolution over several months (4.5 months in soil and 3 months in trees).

We have shown that the stem borehole approach described by Marshall et al. (2020) successfully worked in the field in an angiosperm species in a temperate climate. We adapted the polypropylene membranes inside the stem boreholes and used dried ambient air as carrier gas, reducing the risk of infection and simplifying the field setup.

The upper soil layer was most variable in moisture content and isotopic composition. Spatial heterogeneity was shown by the two soil pits from which one was located under the canopy while the other one was in the open. The canopy-covered soil pit had lower volumetric water contents but more enriched and more variable isotopic composition than the one in open space.

The xylem isotopic composition mostly resembled topsoil water, and mixing models indicated that the soil water source could explain $\sim$ 90 % of RWU. The summer sap flux velocity remained high, suggesting rapid xylem water travel times of a few days, and sap flux velocity showed no obvious response to lower soil water conditions (i.e. reduction) or precipitation events (i.e. increase). At the end of the season, the reliance on upper soil water uptake reduced (but was still $\sim$ 60 %), suggesting use of additional deeper soil waters. It is likely that the willow trees preferred topsoil water in summer due to nutrient supply, while at the end of their growing period they shifted to layers with higher soil water content.

Although our approach delivered novel and reliable data, improvements in terms of setup and choosing locations or time periods with more variable water source availability will benefit further investigations. For long-term in situ xylem water investigations, further research tackling challenges such as night-time tube condensation, compression of the membrane, and testing on different tree species is required.

*Data availability.* The isotope data are available with a password (to be received from the corresponding author upon request) from the open-access database FRED at IGB.

*Supplement.* The supplement related to this article is available online at: https://doi.org/10.5194/hess-26-1-2022-supplement.

*Author contributions.* The study was designed by JL, DT, MD, and CS. Fieldwork and data collection were undertaken by JL and DD. Data were analysed by JL, with ongoing discussion and inputs from DT, CS, MD, and AS. JL prepared the draft manuscript, which all authors subsequently contributed to and edited.

*Competing interests.* The contact author has declared that neither they nor their co-authors have any competing interests.

*Acknowledgements.* The authors are grateful to Jonas Freymüller, Hauke Dämpfling, and Adrian Dahlmann, who were involved in the in situ site setup and maintenance. We also thank Lukas Kleine and Christian Marx for their help in soil and twig sampling and characterisation, as well as Christian Marx for cryogenic extraction of the twig samples. Chris Soulsby's contributions were supported by the Leverhulme Trust ISOLAND project (RPG-2018-375).

*Financial support.* This research has been supported by the BMBF (funding code 033W034A), which supported the stable isotope laboratory, and the Open Access Publication Fund by IGB.

The publication of this article was funded by the Open Access Fund of the Leibniz Association.

*Review statement.* This paper was edited by Laurent Pfister and reviewed by John Marshall and Valentin Couvreur.

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
