# Peer review of "Xylem water in riparian Willow trees (*Salix alba*) reveals shallow sources of root water uptake by *in situ* monitoring of stable water isotopes"

_Hydrology and Earth System Sciences, 2021_

## Referee Comment (RC2)

Review of: "Xylem water in riparian Willow trees (Salix alba) reveals shallow sources of root water uptake by in situ monitoring of stable water isotopes"

This manuscript describes an intensive experiment that monitored the isotopic composition of xylem water in two willow trees and compared it to the available water sources in the soil system. The xylem water most resembled the water in the surface soil, which suggests that most of the water transpired by these trees was from that source. This inference is supported by the drawdown of water content in the surface soil. The paper is one of the first to use the *in-situ* borehole method to monitor rapid changes in xylem water. It was particularly interesting that the traditional cryogenic sampling of xylem water, which has been in serious question for a year or more, yielded isotopic values that neither made sense nor matched the *in situ* samples. Figure 5 can be beautiful--with some modifications.

Two papers describing similar conclusions, but in other species, have appeared in recent months. The first is different because it uses the Volkmann probes, but the second uses boreholes. I am not sure the second can be cited yet, but the first certainly can. These papers are:

Gessler, A., Bächli, L., Rouholahnejad Freund, E., Treydte, K., Schaub, M., Haeni, M., Weiler, M., Seeger, S., Marshall, J., Hug, C., Zweifel, R., Hagedorn, F., Rigling, A., Saurer, M., & Meusburger, K. (2021). Drought reduces water uptake in beech from the drying topsoil, but no compensatory uptake occurs from deeper soil layers. *New Phytologist*, 233:194-206.

and

Kathrin Kühnhammer,Adrian Dahlmann,Alberto Iraheta,Malkin Gerchow,Christian Birkel,John D. Marshall,Matthias Beyer. Accepted. Continuous *in situ* measurements of water stable isotopes in soils, tree trunk and root xylem: field approval. Rapid Comm. Mass Spectrom.

The authors have modified the earlier borehole method by inserting a porous sleeve into the borehole. It is almost certainly not true that fungal infections were prevented by the sleeve (line 224) unless the entire procedure of boring and tube insertion was carried out under sterile conditions. Perhaps it is enough to say that fungal infections were reduced, although even this would seem to require some attempt to examine fungal density on the borehole wall. Second, I am curious about the concern over liquid water (line 222) in the boreholes. As the xylem is under tension, it would seem that this could not occur except perhaps when tension is released as the vessels were first cut. I doubt that the sleeve caused any harm apart from a minor slowing of response, but I am skeptical that it caused any improvement.

Questions about artifacts caused by cryogenic distillation (Chen et al., 2020, cited) are worrying much of the ecohydrology community. As this study has compared this technique to the borehole technique, confirming that the worries are valid, I think that the comparison should be emphasized. I suggest presenting it in the abstract (not corroboration, line 9) and the conclusions, for example. It is relevant as it helps justify preference for some version of the borehole technique in future work.

In the methods, you should probably add measurements of height and diameter of the trees.

I found the order of the figures confusing. Fig. 7 should be presented after Fig. 3 as these are similar in structure and both present something like raw data. Fig. 4 can be moved to the supplement as it is not really used or discussed here. Fig. 5 could then come next, after being modified somewhat. In Fig 5, the polygons obscure the color of the points, making it difficult to distinguish which points are which. Especially difficult is to see the surface soil data in the green and blue polygons. I suggest colored outlines rather than filled polygons. Also, there is a series of precip data in the middle of the plot that almost look too perfect. Perhaps those could be checked and commented on in the text? This figure is otherwise compelling. I look forward to seeing its final version.

Fig. 6 would come next in logical order as lc-excess is derived from the raw isotopes, but the caption should include a brief description of what lc excess is and the labels on the axes should be improved (lc ex. is not OK). Finally I am curious about why the precip lc excess in fig. 6c is so far above zero. It should be zero by definition, should it not?

Finally, two comments on Fig. 9. There is no dashed blue line in the legend. In addition, much of the soil data is cut off at the top and bottom of the plots. This would be more compelling if they were included.

In the discussion before line 381, probably at the end of the previous paragraph, I would note that the dynamics of vwc are also consistent with the conclusion that most water uptake is from the upper layer. I presume that there are few data points above field capacity, so the seasonal pattern of vwc should be driven primarily by root uptake. By the way, this raises the question of what you have actually gained by the isotopic measurements if vwc shows the same thing. I think there are good answers, but it would be worthwhile to spend a few lines on this question near the end of the discussion.

---

## Author Response (AR2)

**Comments to the author**:
Dear authors,
following up on the assessments made by the referees and your responses during the discussion phase, I invite you to submit a revised version of your contribution. As already stated in your replies to the comments made by the referees, please consider in particular the following aspects:
- further emphasizing on the comparison between cryogenic distillation and the borehole technique in the light of artifacts related to cryogenic extraction;
- reconsidering the discussion of potential implications inherent to the comparison of soil water content and in-situ monitoring of water signatures;
- reconsidering your conclusions/statements on the production of reliable 2-hourly resolution data for water signatures.
Please make sure to also take into consideration the other points listed by the referees (e.g., related to the figures [already dealt with during the discussion phase]).
I am looking forward to receiving the revised version of your manuscript!
Best regards,
Laurent Pfister

Response to the Editor:
Dear Dr. Pfister,
We would like to thank you for the opportunity to revise our manuscript and thank the reviewers for the constructive suggestions and comments. We addressed all comments – our responses are detailed below. These revisions include in particular: updated figures with minimum 300 dpi, referencing to the recent findings from Gessler et al. (2021) and Kühnhammer et al. (2022), we revised the discussion about the ecohydrological parameters, we shortened the discussion on high-resolution data and revised the method section in terms of lc-excess and the borehole equilibrium method. Last but not least, we stronger emphasized the importance of *in-situ* measurements of stable water isotopes as a tool to investigate root water uptake.
The authors believe that this revision has strengthened this manuscript clearer and hopefully, it can now be accepted for publication.
Best Regards,
Jessica Landgraf on behalf of all coauthors

**Comments to the author**:
Dear colleagues,
after having assessed your revised manuscript, I can confirm that the main points that had been identified by the referees have been properly addressed.
Your manuscript can now be accepted for publication in HESS and will go through the final processing steps.
Congratulations on this very nice achievement. Thank you for having chosen HESS for publishing your research.
Best regards,
Laurent Pfister

Response Editor:
Dear Dr. Pfister,
On behalf of the authors, we thank the editor for accepting the revised manuscript for publication in HESS. We are happy to support the processing steps and are looking forward to publication.
Best Regards,
Jessica Landgraf on behalf of all coauthors

*Reviewer 1*
*In this study, the authors investigate the sources of water absorbed by Willow trees using both bulk and in situ monitoring of water isotopic signature in the soil and stems of two of these trees. A Bayesian mixing modelling approach to water sourcing suggests that the vast majority of water is absorbed from the top soil (above 40 cm depth), except at the end of the summer period when a fraction of the xylem water could originate from deeper soil layers (40 cm and 100 cm).*

*General comments:*
*From a general point of view, I believe the manuscript is sound, well-written and structured. It is also very well adapted to the audience of Hydrology and Earth System Sciences. On the one hand, I found the comparison of soil water content and in situ monitoring of water signatures particularly interesting and think the discussion could emphasize the related potential implications a bit more. On the other hand, the conclusion on the production of reliable 2-hourly resolution data for water signatures I think would need further support in the main text or should be tuned down. My main concern is about the quality of the figures, which would require substantial work. Yet, I think the manuscript is overall in a good shape and could be considered for publication in HESS after minor revisions.*

**Response to General Comments**
The authors thank Dr. Couvreur for his constructive comments. In the revision, we emphasized more strongly the relationship of in-situ and other ecohydrological parameters like soil water content in the discussion. In turn, the discussion about the reliability of 2-hourly resolution data was shortened. All figures were uploaded with a higher resolution (300 dpi) and larger axis labels to ensure high quality.

We added all the revised figures into this document here below (which addressed all the constructive suggestions by the reviewer) and will upload them separately at high resolution.

*Specific comments:*
*Lines 81-84 (L81-84): Besides radial growth, doesn't the swelling / shrinking response of tree stem diameter reflect stem water tension, which results from the imbalance between canopy transpiration and root water uptake? This kind of formulation would seem more intuitive to me. Was any such swelling / shrinking fluctuation observed daily? If not, this part of the introduction could as well be removed to make the manuscript more concise.*
**Response to L81-84:** The authors thank the reviewer for their suggestion. Yes, there was daily swelling / shrinking observed in the tree stems of the willows. The authors have now emphasized the daily cycle of shrinking and swelling more clearly in the results section. And yes the reviewer is correct: so we also stress that the swelling / shrinking reflects stem water tension.

*L85: I think this part needs a bit of rephrasing too as transpiration is considered as the product of VPD and canopy conductance (often termed gs). Canopy conductance is considered to react to multiple factors like limited light, limited export of products of photosynthesis and limited water availability. Therefore, I think writing that sap flow usually reflects VPD is a bit strong. Maybe clarify "in absence of (...) limitations (...)".*
**Response to L85:** Thank you for pointing this out. The authors clarified the relationship of sap flow and VPD during the revision.

*L88-89: Here I would suggest to re-order the terms. It could be more intuitive to mention decreased sap flow after low leaf water potential, turgor, and stomata closing, as it would come last in a temporal sequence of observations.*
**Response to L88-89:** Great suggestion. The authors changed the order accordingly.

***Figure 1:*** *There seems to be something wrong with the coordinates along the frame of panel A. Could you check if that is the case? Also, several elements in the figure are very small and hard to read. For instance, the labels in panel B, the dots marking soil pits and probes, whose colours are hard to distinguish for such small points. Could you modify the figure to facilitate its readability?*
**Response to Figure 1:** The coordinates along the frame were set automatically by using QGIS. The coordinate systems is World Geodetic System 1984 used in GPS (EPSG:4326 - WGS 84) and was updated to Universal Transverse Mercator (UTM). The figure was modified to improve its readability.

***L143-144:*** *Can the eddy flux covariance measurement be considered as representative of the Willow trees transpiration? If not, could you clarify already at this point how the data is intended to be used?*
**Response to L143-144:** No. For this the footprint of the Eddy flux is too closer to ground level. However, important to note is that the measured ET data were used as a normalized time series in terms the variability in the ET dynamics rather than the absolute values of Willow transpiration. In revision, the authors clarified the usage of Eddy Flux data in terms of using the Eddy flux data as an indicator of dynamics / variability in ET (rather than absolute values).

***Figure 3 (and others):*** *The use of the symbol "/" preceding units in the y-axis label is a bit confusing. Could you use another symbol, like the pair of brackets for instance?*
**Response to Figure 3 (and others):** We used the axis description after ISO/IEC 80000. However, for improved readability we changed this for all units in the revision.

***Figure 4:*** *The right-side y-label would be clearer specifying "stem diameter variation" as it seems the stem size was not close to zero in May 2020. Northern and Southern Willow labels directly in the Figure would also be convenient. The legend indicates "Sap flow N" in both panels but it is unclear what "N" stands for. Could you clarify? Sap flow rates and ET would also gain at being overlain all in panel 4a for easier comparison, while stem diameter variation could be displayed in panel 4b.*
**Response to Figure 4:** Thanks for the comments. We updated figure 4 accordingly. There was a typo causing the beginning of the measurement to be above 0 for the stem diameter variation. This has been corrected for in the updated version of the Figure. Further, "Sap flow N" was describing the sap flow measurement from the northern site of the stem. In the revised version this description is no longer used while the measurement position was clarified in the content.

***L290:*** *The subtitle could also mention "and transpiration".*
**Response to L290:** The authors updated the subtitle.

***Figure 5:*** *The figure is particularly hard to read as symbols are small, several of them have similar colours and/or are hidden under geometrical elements. If the ordering of the legend is conserved in the bar plots, it is confusing that precipitation is associated to light-grey bars but black dots, surface water medium-grey bars but white dots, etc. Could you work on a new version of this figure that is easier to read?*
**Response to Figure 5:** This was a good suggestion. We changed the plot to only show polygon-outlines and precipitation, surface water, and groundwater were plotted with dots matching the color of their bar plot counterparts.

***L307:*** *Here and at other places in the manuscript, the use of the "-" sign to specify ranges of values while these values are negative is not ideal. Could you replace the "-" sign by the word "to"?*
**Response to L307:** This is a good suggestion. The authors replaced "en dashes" used for ranges by "to" in the revised manuscript.

***L309:*** *I am relatively new to water isotopic studies, so the concept of "lc-excess" I had to look up. I think for the audience of HESS it would be worth carefully defining the lc-excess with the associated equation and possibly a one-sentence example to make it easier to grasp.*

**Response to L309:** The authors thank the reviewer for this comment and clarified lc-excess in the methods section including the equation.

*L312: The expression "generally similar to" is a bit vague given the wealth of data available in this study. Were in situ and bulk measurements significantly different? Is there any indicator of their similarity (e.g. R-square, …)?*
**Response to L312:** We now conducted a similarity analysis with Euclidean distance, the values are given now.

*L314: Please clarify the type of variability (space, time, …).*
**Response to L314:** The authors updated this section by stating that the higher variability of Pit A compared to Pit B over time is caused by the spatial differences of the two soil pits.

*L339: Could you comment on what could make responses more marked in covered pit A than pit B?*
**Response to L339:** Pit A was directly under one of the Willow trees and higher interception seems to have resulted in much drier soil conditions at Pit A compared to Pit B. Thus, Pit A relatively to B had smaller water amount present in the soil. Due to these generally smaller water amounts incoming precipitation varied the isotopic composition of the top soil water of Pit A more intense than it did at Pit B. We clarified this section in the revision.

*Figure 7: In panel (a), the location of the "zero" differs between vertical axes, which complicates the visualization of the results. Could you fix this? Some of the panels (b) to (e) could as well be merged to facilitate the comparison of the results and the visualization overall as vertical axes are currently very small.*
**Response to Figure 7:** Thank you for pointing this out. The authors reorganized the zero-lines from panel (a) x-axis on one line. Further, panel (b) and (c) as well as (d) and (e) were merged to one graph.

*Figure 9: Several of the box plots are hard to distinguish from the border of the panels. Could you make them easier to distinguish for instance with a lighter panel border colour? The legend for soil moisture lacks the dotted / dashed aspects. Could you make the legend more consistent with the content of the figure?*
**Response to Figure 9:** These are good suggestions. We improved the clarity of the box plots. Further, the legend was extended to also include the dotted and dashed line.

*L358-359: The prediction of water uptake almost solely from the top soil in the Bayesian modelling output is quite interesting. If you had to explain the remaining differences between water isotopic signatures in the stem and at 10 cm depth, what would be the other sources that you would consider as necessary complements? Could you discuss the results also from this perspective? From a quantitative point of view, I think the authors could as well have argued (but don't have to) on the possibility to have root water uptake solely from the top 40 cm of soil by comparing the cumulative tree transpiration and rainfall to the soil water storage change. My estimations suggest that, even when neglecting water capillary rise from deeper layers, the water balance in the top soil seems reasonable if the tree roots extend about 10 meters away from the stem.*
**Response to L358-359:** Thank you for these suggestions. With the setup of the in-situ site we considered groundwater, lake, stream as well as soil water to be the main sources of root water uptake. The groundwater, due to the adjacent lake, is relatively shallow with ~2 m below soil surface and well mixed with the lake water. However, the reviewer is correct, in terms of extensive root networks in the upper soil. Since this paper was submitted, a companion paper used the data presented here in a process-based ecohydrological model (which resolves the water and isotope mass balances) demonstrated this. We now refer to this in the revised manuscript to clarify this issue.

**L376:** *I found the phrasing "soil moisture availability was (…) not less than (…)" a bit odd. Do you mean that it is higher (as suggested by Fig. 9) but not significantly?*
**Response to L376:** The soil moisture in September was not significantly different to hotter summer month like July and August. We clarified this in the revision.

**L394-397:** *This part of the discussion is quite interesting. In the results I kept wondering why deeper water uptake occurred just as the upper soil was rewetting, which is counterintuitive for a specialist of hydrodynamics, as passive water flow from shallower soil layers to roots is supposed to increase as the water potential in these layers increases (a process called "root water uptake compensation"). So, your observations seem to go against the second law of thermodynamics (how exciting!). You do mention the possible explanation that the stem water signature could be the integral of past and present water uptake (thereby possibly reconciliating your observations with thermodynamics laws). If that is the case and if the volume of the stem water pool mixing with newly absorbed water is large enough, one would expect to see the signature of deep water in stems weeks after deep water uptake occurred. Obviously, this calls for more investigations of the mixing of soil and stem water pools ideally under controlled conditions and with labelled water, as well as for new versions of Bayesian mixing models that account for such a temporal integration of water signatures (or do they exist already?). The temporal integration of water signatures would also alter the ability to infer on water uptake profiles at high-temporal resolution, unless one was able to "deconvolute" the temporal dynamics of stem water signatures. Could you discuss this a bit more in depth in the manuscript?*
**Response to L394-397:** Thank you for these constructive comments. It's true that our assumption seems inconsistent with the second law of thermodynamics, but since trees are living organisms, it's possible they found a way around this problem at least in the short term *(like giraffes found a way to breath with an air tube volume larger than the one of their lungs… physically impossible)*. We thought this being a possibility, because other studies found willows to prefer water from drier soil parts even though it could be accessed more easily in stream or groundwater (cf. Martilla et al. 2017). If biological activity is behind this phenomenon, it will be fascinating to identify the physical/biological processes behind it. However, other processes like mixture of "new" and "older" water in the xylem could explain our results as well and are already discussed in detail the study from Mennekes et al. (2021). We agree that further condition-controlled experiments are required to determine the processes behind our results in detail. Further, a mixing model addressing the temporal integration is also favorable for stable water isotope investigations in xylem water. Moving-window analysis might be an easy but very useful approach to evaluate the temporal variability. We revised this in further detail.

**L418-440:** *Here diurnal fluctuations of the signals are discussed in depth, which I found a bit odd as I did not find supporting data and results in the main text and figures (I guess they are in appendices). I think if they are to be discussed in depth, they should be presented in the results and figures for the audience to have a grasp on what was observed at such a temporal resolution. It is striking in particular to conclude (L453-454) on the production of "reliable high-frequency (…) 2-hour resolution" observations while I think only daily-averaged data is presented in the body of the manuscript. I think it would be more consistent to present the high-resolution data and demonstrate confidence in its quality, or not to conclude on the production of such high-resolution reliable data.*
**Response to L418-440:** You are correct. We reduced the discussion about sub-daily data (see also our response to your comment further above).

**L463-464:** *This sentence is unclear to me. Could you rephrase it?*
**Response to L463-464:** The authors clarified that ~90% of root water uptake was estimated by the mixing model to be linked to the first 10 cm of the soil.

**Typos:**
**L25:** *I think the expression "uptaking" is incorrect and the correct version is "taking up".*
**Response to L25:** The authors used the correct term of "taking up".

*L326: The signs "?:" seem to be written in place of " for".*
**Response to L326:** The authors removed the signs "?:" from the text.

*L413: I think N2 requires a subscript for the number 2.*
**Response to L413:** This was corrected.

**Reviewer 2**
**General Comments:**
*"Xylem water in riparian Willow trees (Salix alba) reveals shallow sources of root water uptake by in situ monitoring of stable water isotopes" This manuscript describes an intensive experiment that monitored the isotopic composition of xylem water in two willow trees and compared it to the available water sources in the soil system. The xylem water most resembled the water in the surface soil, which suggests that most of the water transpired by these trees was from that source. This inference is supported by the drawdown of water content in the surface soil. The paper is one of the first to use the in-situ borehole method to monitor rapid changes in xylem water. It was particularly interesting that the traditional cryogenic sampling of xylem water, which has been in serious question for a year or more, yielded isotopic values that neither made sense nor matched the in situ samples. Figure 5 can be beautiful--with some modifications.*

**Response to General Comments**
The authors thank Prof Marshall for his positive comments and for acknowledging the data-rich nature of our experiment, but also the novelty of this paper using the in-situ bore hole method (developed by himself).
We emphasized the usage of the borehole method as being one of the first. Further, we discussed the cryogenic results of xylem water in more detail in the revision. Figure 5 was modified to be more readable.

**Additional literature:**
*Two papers describing similar conclusions, but in other species, have appeared in recent months. The first is different because it uses the Volkmann probes, but the second uses boreholes. I am not sure the second can be cited yet, but the first certainly can. These papers are:*

*Gessler, A., Bächli, L., Rouholahnejad Freund, E., Treydte, K., Schaub, M., Haeni, M., Weiler, M., Seeger, S., Marshall, J., Hug, C., Zweifel, R., Hagedorn, F., Rigling, A., Saurer, M., & Meusburger, K. (2021). Drought reduces water uptake in beech from the drying topsoil, but no compensatory uptake occurs from deeper soil layers. New Phytologist, 233:194-206.*
*and*
*Kathrin Kühnhammer,Adrian Dahlmann,Alberto Iraheta,Malkin Gerchow,Christian Birkel,John D. Marshall,Matthias Beyer. Accepted. Continuous in situ measurements of water stable isotopes in soils, tree trunk and root xylem: field approval. Rapid Comm. Mass Spectrom.*

**Response to additional literature:** The authors thank the reviewer for their suggestion. We added the papers to our discussion and firmly compared our results with their recent findings. The second paper is now published, so both studies were referred to and discussed in our revision.

**Method:**
*The authors have modified the earlier borehole method by inserting a porous sleeve into the borehole. It is almost certainly not true that fungal infections were prevented by the sleeve (line 224) unless the entire procedure of boring and tube insertion was carried out under sterile conditions. Perhaps it is enough to say that fungal infections were reduced, although*

*even this would seem to require some attempt to examine fungal density on the borehole wall. Second, I am curious about the concern over liquid water (line 222) in the boreholes. As the xylem is under tension, it would seem that this could not occur except perhaps when tension is released as the vessels were first cut. I doubt that the sleeve caused any harm apart from a minor slowing of response, but I am skeptical that it caused any improvement. Questions about artifacts caused by cryogenic distillation (Chen et al., 2020, cited) are worrying much of the ecohydrology community. As this study has compared this technique to the borehole technique, confirming that the worries are valid, I think that the comparison should be emphasized. I suggest presenting it in the abstract (not corroboration, line 9) and the conclusions, for example. It is relevant as it helps justify preference for some version of the borehole technique in future work. In the methods, you should probably add measurements of height and diameter of the trees.*

**Response to Method:** The authors thank the reviewer for their constructive comments. We agree with the reviewer, that an active prevention of fungal infection by the polypropylene membranes inside the boreholes will require further investigation. Due to the small pores of the membrane, which are impermeable to fungal spores, we think that a "reduced risk of fungal infection" can be assumed without testing. We clarified this in the revision. Since the membrane usage was new, we wanted to emphasize every possible improvement of the setup. We did not expect the willows to have liquid water inside the borehole, but as the reviewer knows, there are (tropical) tree species, that use water inside wounds as a defense mechanism. Under such circumstances the membranes would only allow vapor to enter the system. However, since this is not the case in our study, we removed this section in the revision. It might not be a direct improvement of the borehole method, but we think it's still an advantage that the same method (membranes inserted inside a matrix) can be used inside soil and xylem to collect the sample improving comparability of the two measurements.

We thank the reviewer for their suggestion and compared in-situ and cryogenic extraction of xylem water in more detail also emphasizing the results in the abstract.

Further, estimations of tree height and measurements of diameter of the trees were stated in the revision.

**Order of figures:**

*I found the order of the figures confusing. Fig. 7 should be presented after Fig. 3 as these are similar in structure and both present something like raw data. Fig. 4 can be moved to the supplement as it is not really used or discussed here. Fig. 5 could then come next, after being modified somewhat. In Fig 5, the polygons obscure the color of the points, making it difficult to distinguish which points are which. Especially difficult is to see the surface soil data in the green and blue polygons. I suggest colored outlines rather than filled polygons. Also, there is a series of precip data in the middle of the plot that almost look too perfect. Perhaps those could be checked and commented on in the text? This figure is otherwise compelling. I look forward to seeing its final version.*

**Response to order of figures:** These are good suggestions. We reordered the figures to better match the processing of the data. Further, we updated Figure 5 to improve its readability by using unfilled polygons. We double checked the precipitation data points matching the local meteoric waterline and didn't find any anomalies (sample amount between 10 and 500 ml, results are from June or July, sometimes due to the 4h interval several measurements were aggregated with weighted mean to get daily results). Precipitation events were also compared with data from the German weather service (DWD). We assume the data match the local meteoric waterline since the line is calculated from those data (from May 2020 – January 2021).

*Fig. 6 would come next in logical order as lc-excess is derived from the raw isotopes, but the caption should include a brief description of what lc excess is and the labels on the axes should be improved (lc ex. is not OK). Finally I am curious about why the precip lc excess in fig. 6c is so far above zero. It should be zero by definition, should it not?*

**Response to Fig. 6:** Since this was also suggested by reviewer 1, we added a section in the methods part about lc-excess, explaining the value and giving the calculated function we used. Further, we added a short description in the figure description of figure 6. The positive lc-excess was produced by a typo reversing + and - in the calculation of precip, groundwater, lake and stream water lc-excess. We corrected this in the revision.

*Fig. 9:*

*Finally, two comments on Fig. 9. There is no dashed blue line in the legend. In addition, much of the soil data is cut off at the top and bottom of the plots. This would be more compelling if they were included.*

**Response to Fig. 9:** Thanks for pointing this out. We updated figure 9 to include the description of the dashed lines and the cut off parts of the soil data at the top and base of the plots.

*Discussion:*

*In the discussion before line 381, probably at the end of the previous paragraph, I would note that the dynamics of vwc are also consistent with the conclusion that most water uptake is from the upper layer. I presume that there are few data points above field capacity, so the seasonal pattern of vwc should be driven primarily by root uptake. By the way, this raises the question of what you have actually gained by the isotopic measurements if vwc shows the same thing. I think there are good answers, but it would be worthwhile to spend a few lines on this question near the end of the discussion.*

**Response to Discussion:** Good point! It's true that vwc of the top soil is probably mainly driven by root water uptake, but it's also important to consider the other soil layers. The first Reviewer mentioned that from a quantitative perspective the willows could easily have used mainly the water from 40 cm instead of the water at 10 cm. Hence, we see here the importance of the stable water isotope analysis linking the root water uptake mainly to the top soil. In general, we think that's a very interesting point to discuss and we emphasized this in more detail in our revision.

[Figure]

Figure 3. Hydroclimatic conditions showing daily precipitation (a), air temperature (b), vapor pressure deficit (c), evapotranspiration (d), and soil moisture for Pit A (e) and Pit B (f).

[Figure]

Figure 4. Daily total sap flow measured at the northern sap flow sensor and stem diameter variation measured for the stem-radius. Plot a) shows the results of sap flow together with Evapotranspiration for comparison and b) the results of the stem diameter variation during the measuring period.

[Figure]

Figure 5. Dual isotope plot of in situ (daily) soil and xylem as well as precipitation (daily), surface and groundwater (weekly) sampling. Soil and tree data are highlighted with boundary polygons for 10 cm, 40 cm, 100 cm and tree (upper and lower results joined) clusters. Additional boxplots show the sample distribution of the data sets.

[Figure]

Figure 6. Box plots showing the isotopic composition of daily precipitation sampling, weekly sampled groundwater, lake and stream water, and in situ sampled soil, and xylem water.

[Figure]

Figure 7. In situ time series of daily δ2H in: Precipitation (a), Pit A and Pit B (b), and Northern Willow and Southern Willow (c).

[Figure]

Figure 8. In situ time series of daily lc-excess in: Precipitation (a), Pit A and Pit B (b), and Northern Willow and Southern Willow (c).

[Figure]

Figure 9. Estimated percentage of willow water from each soil depth in (a) Northern Willow and (b) Southern Willow. The box plots for each source show the 25th and 75th percentiles, with whiskers extending to the minimum and maximum estimated proportion. Gray and white bands show the division of each week. The right-side y-axis shows the soil moisture for each soil depth (weekly average).